# Modeling the detection range of pulsed calls from resident killer whale in nearshore waters of British Columbia, Canada

Xavier Mouy[1,¤,*], Melanie Austin[1], Jennifer Wladichuk[1], Harald Yurk[2]

**1** JASCO Applied Sciences, Victoria, British Columbia, Canada, **2** Pacific Science Enterprise Centre, Fisheries and Oceans Canada, West Vancouver, British Columbia, Canada

☯ These authors contributed equally to this work.
¤ Current Address: Applied Ocean Physics and Engineering department, Woods Hole Oceanographic Institution, Woods Hole, MA, USA
* xavier.mouy@whoi.edu

## Abstract

Passive acoustic monitoring (PAM) using underwater listening stations is widely employed to assess the presence and movements of marine mammals. Accurate interpretation of PAM data requires knowledge of vocalization detection ranges, which vary spatially and temporally with ambient sound levels and sound propagation conditions. This study presents a Monte Carlo framework for estimating call detection probabilities as a function of distance incorporating variability in frequency dependent source levels, ambient sound levels and caller depths. This methodology was applied to underwater listening stations deployed by Fisheries and Oceans Canada in the Salish Sea to monitor endangered Southern and threatened Northern Resident Killer Whales (*Orcinus orca ater*). The approach integrates *in situ* ambient sound measurements and modeled propagation losses to account for variability in source levels and vocalizing depths. To reflect frequency-dependent detectability by an automated detector, the analysis was performed independently across consecutive 300 Hz frequency bands. Median estimated detection ranges for Southern Resident Killer Whale pulsed calls varied from 650 m under the worst conditions (high ambient noise levels, low source level, high propagation loss) to 7.9 km under the best conditions (low ambient noise levels, high source level, low propagation loss). Maximum detection ranges were generally greater in summer than in winter, primarily due to higher ambient noise levels in winter associated with increased weather activity. Calls from Northern Resident Killer Whales were detectable at shorter ranges than those from Southern Residents, reflecting their lower source levels and weaker low-frequency components. Sensitivity analysis showed that the frequency distribution of source levels was the primary factor influencing detection range estimates, while seasonal changes in propagation loss had comparatively limited impact. Although developed for killer whales, this approach can be adapted to other vocal species to quantify

**Data availability statement:** All SPL and propagation loss coefficients files are available from the OSF repository located here: https://osf.io/6ctjq/?view_only=f4e6e83b9eab427b-b530a74638ba3e52.

**Funding:** The authors are part of the organizations JASCO Applied Sciences and Fisheries and Oceans Canada which funded this work and consequently participated to the study design, analysis, preparation of the manuscript, and decision to publish. The Woods Hole Oceanographic Institution had no role in study design, data collection and analysis, decision to publish, or preparation of the manuscript.

**Competing interests:** The authors have declared that no competing interests exist.

species-specific detection range probabilities, guide optimal hydrophone placement to maximize coverage in noisy environments (with application to noise impact mitigation strategies), and provide needed inputs for passive acoustic density estimation models.

## Introduction

An important question in conservation biology, beyond understanding the factors that determine the presence of animals in a particular environment, is how much time they spend within suitable habitats, and how they move within and between them. Addressing these questions can improve our understanding of animal movements within and between habitats, assess the impact of human disturbances, and guide the development of effective mitigation and avoidance strategies [1,2]. However, detecting and tracking movements in a marine environment, particularly for highly mobile animals such as whales and dolphins, remains a significant challenge [3]. Reliable detection of whales and the timely transmission of this information to monitoring stations is crucial when the goal is to mitigate the adverse effects of human activities [4,5]. For instance, such detection is essential for alerting intervention teams to implement mitigation measures before whales enter high-risk areas, such as those affected by oil spills, areas surrounding loud sound-producing activities, or the path of a moving vessel.

Fish-eating *resident* killer whales (*Orcinus orca ater*) in British Columbia, Canada, are primarily represented by two distinct populations: the Southern Resident Killer Whale (SRKW) and the Northern Resident Killer Whale (NRKW). Both populations are known for their foraging preference on Chinook salmon *(Oncorhynchus tshawytscha)* [6]. The SRKW population, which primarily inhabits the waters off the coast of Washington State to the waters north of Vancouver Island is listed as endangered, with currently only 74 individuals remaining [7]. The NRKW population, which primarily inhabits the waters off Washington State up to Glacier Bay in Southeast Alaska has been increasing in numbers since 1973 (est. 340–348 in 2022) but are still listed as threatened [8]. Both populations occur in parts of the inshore waters of the Salish Sea, which ranges from Puget Sound in the South to the Northern sections of the Strait of Georgia to the western entrance of Juan de Fuca Strait (Fig 1). The Salish Sea is also characterized by high levels of human activity on the water, including both small and large commercial vessel traffic, commercial and recreational fishing, whale and wildlife watching, and recreational boating. It has a higher density of human settlements than any other coastal area of British Columbia [9]. SRKW face significant threats from reduced prey availability, environmental contamination, and vessel-based acoustic and physical disturbances [10]. The distribution of both populations, however, appears to be influenced by prey availability and environmental conditions [11–16]. For example, certain areas of the Salish Sea experience windy conditions even during summer resulting in elevated underwater ambient sound levels compared to other areas [17], which have been suggested to affect SRKW

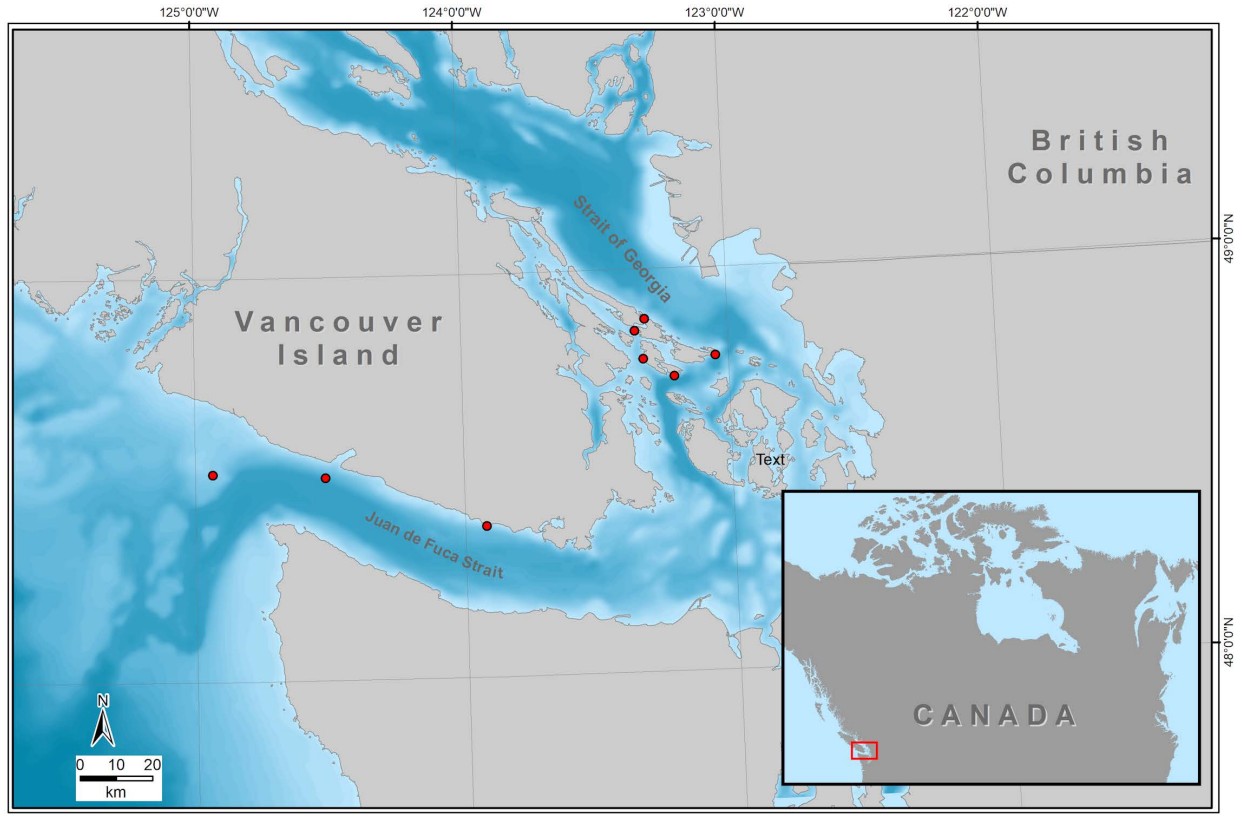

**Fig 1. Map of the study area indicating locations in and adjacent to the Salish Sea in British Columbia, Canada where call detectability of killer whale calls was investigated.** Red dots indicate recorder locations.

distribution and behavior [16]. Effective monitoring and conservation efforts are crucial to support the recovery of these endangered whales [10,18].

Passive Acoustic Monitoring (PAM) is a powerful tool to detect vocalizing cetaceans and has been used effectively to determine habitat presence, spatial and temporal use of habitats, and movements of animals (e.g., [5,19–22]). Its effectiveness has substantially increased in the last few decades due to the development of digital acoustic recording devices that can collect, store, and transmit large amounts of acoustic data [21]. This technological advancement has enabled continuous and long-term acoustic monitoring of underwater environments. Compared to other monitoring methods such as visual observations, PAM has several advantages including that it can observe animals while they are underwater and is less affected by daylight, weather, and sea conditions [23]. However, it is only useful for monitoring animals that produce sounds [24]. Killer whales are highly soniferous which makes them good candidates for PAM. They produce a variety of sounds including echolocation clicks during foraging and navigation, as well as narrowband whistles and broadband pulsed calls used in social communication. Discrete pulsed calls are the most commonly heard vocalizations that occur in all behavioral contexts and can be distinguished aurally and sorted into distinct repertoires identifying different populations and social groups within populations [25]. Clicks are short pulses that usually occur in a series with varying repetition intervals. A click duration ranges from 0.1 to 25 ms [26]. Whistles are single narrow-band tones with little harmonic structure in the 1.5–18 kHz frequency band. Their duration ranges from 50 ms to 12 s [26]. Pulsed calls are harmonically structured with frequency ranges between 80 Hz and 40 kHz [27,28]. Numerous studies have shown that PAM is an effective tool for

enhancing monitoring efforts of killer whale habitat use, particularly during periods when visual observation is challenging, such as at night and during the winter months [29–40].

As part of the Recovery Plan for the endangered SRKWs [10], the Government of Canada's Department of Fisheries and Oceans (DFO) installed PAM stations located in the inland waters around the southern Gulf Islands and the southwest coast of Vancouver Island in British Columbia, Canada (Fig 1). Some of these stations stream data to a central location where automated whale call detection and classification tools process the data. The goal of this network is to provide near real-time detection and tracking of killer whales, specifically SRKW, and send alerts about potential threats to the whales, such as the possibility of physical disturbance from whale-vessel interactions and potential vessel strikes.

To properly interpret the results from PAM networks for the purpose of whale movement tracking, it is important to estimate the distance over which the signals of interest can be detected. This detection range is influenced by the ambient sound in the environment, the sound propagation, and the properties of the sounds we want to detect. Because these factors are not static, the detection range can be highly dynamic and may vary among locations and seasons [41]. Ambient sound fluctuates with weather conditions (e.g., wind, waves) and anthropogenic activities (e.g., vessel traffic). Sound propagation is influenced by environmental factors, such as bathymetry, water temperature, and pressure which affect the sound speed profile [42]. Whale calls and acoustic behavior have evolved in these dynamic soundscapes which have led to adapted acoustic signaling. Commonly, propagation loss (PL) modeling is used to estimate detection ranges [43–45]. In most cases in which PL modeling was applied, environmental conditions influencing the detectability of calls have been assumed to be somewhat constant, and averages were used in the model assumptions. In more recent years, environmental conditions are seen as dynamic influences on the detectability of whale calls [46] and ambient sound variation has been considered an important influence on the detectability of calls [47]. To improve the interpretability of PAM studies, it is important to recognize the association of influences such as ambient sound variation and the distribution of energy across the signal frequency spectrum on the detectability of calls. Most published cetacean acoustic detection range studies focus on baleen whale calls, which, except for species that produce elaborate songs (e.g., humpback and bowhead whales), typically concentrate the majority of their call energy within a narrow frequency band of a few tens of hertz (e.g., blue, fin, right whales; [44,48]). This tends to make it easier to measure source levels of calls and determine propagation loss in the narrow frequency band in which the call energy is located. In contrast, many odontocete calls (e.g., killer and pilot whale calls, beluga whale calls and burst pulses of smaller dolphins) and humpback whale songs are broadband signals whose energy is spread out over a wider frequency band (several thousand hertz). It appears more difficult to determine detection ranges of broadband vocalizations due to the more complex frequency dependence of propagation losses, source levels, and interfering noise levels.

The objective of this study is to estimate the detection range of individual SRKW and NRKW calls at several locations of the DFO PAM stations over time and evaluate their ability to detect killer whales and their movements through the Salish Sea. This effort focused solely on the detection of discrete pulsed calls since they occur in all behavioral contexts and travel further than whistles and clicks. To estimate the influence of the dynamic physical acoustic environment on call detectability, call detection range variation was modeled at eight PAM locations. A probabilistic modeling approach was implemented using randomized inputs of call source levels and ambient sound levels across dominant frequency bands. Detection ranges were estimated by applying modeled PL, accounting for source depth variability based on a depth distribution from killer whale tagging data. This allowed for predictions of received levels at each of the eight PAM stations.

## Methods

### Approach

The acoustic detection range of a killer whale call was estimated by calculating the distance from a bottom mounted hydrophone where the received sound level of a call remains more than a detection threshold (DT) above the ambient

sound level in the same frequency band. The received sound level of a killer whale call (RL) is the difference between the sound level at the source (i.e., the whale) and the propagation loss between the whale and the hydrophone [49]:

$$RL \; = \; SL - PL \,, \tag{1}$$

where SL is the source level in dB re 1 µPa$^2$m$^2$ and PL is the propagation loss in dB re 1 m$^2$. For this analysis, we obtained killer whale call source levels from published and unpublished literature and ambient sound levels recorded at (or near) the model locations (Table 1). We calculated PL using a specialized acoustic model that integrates parabolic equation and ray tracing methods (MONM; JASCO Applied Sciences, Ltd.) and accounts for source depth, receiver depth, environmental and bathymetric conditions between the source and receiver, and absorption. The modeled propagation loss values were plotted as a function of range and the data were fit with an equation of the form:

$$PL(f, z, R) = A(f, z) - n(f, z)log_{10}R \tag{2}$$

for frequency ($f$) in Hertz, depth ($z$) in meters, and range ($R$) in meters. For interpretation purposes, the parameter $n$ in Eq 2 is representative of a geometric spreading loss term and characterizes the amount by which the sound level decreases with the logarithm of distance away from the source for each modeled location. The maximum distance at which a call can be detected is where the call's received level, RL, exceeds the ambient sound level, NL, at the recorder in the same frequency band by at least the detection threshold, DT:

$$RL(f, z, r) \geq \; NL(f) + DT, \tag{3}$$

DT was set to 5 dB for this analysis, as it was empirically shown to be the signal-to-noise ratio above which automatic killer whale call detectors perform reliably [50]. The detection threshold used here strictly represents the signal processing detection threshold for automated detectors and is not related to the listening detection threshold of the animals.

At a given source depth, the detection range was estimated separately for each 300 Hz frequency band between 1000 and 14800 Hz (46 frequency bands total) and was calculated as:

$$R(f) \; = \; 10^{\frac{A(f) + SL(f) \; - \; NL(f) \; - \; DT}{n(f)}} \tag{4}$$

The final maximum detection range was then defined as:

$$R_{max} \; = \; \mathrm{argmax}_f \; (R(f)) \tag{5}$$

**Table 1. Acoustic source levels of killer whale calls from relevant literature for different call types and populations of killer whales in this study.**

| Study Area | Population | Call type(s) | Source levels in dB re 1 µPa @ 1 m (mean±SD) | Reference |
|---|---|---|---|---|
| Johnstone Strait, BC | NRKW | Stereotyped calls | 152.6±5.9 | Miller [27] |
| Johnstone Strait, BC | NRKW | Stereotyped calls | 145.3±6.8 | Wladichuk et al. [55] |
| San Juan Islands, WA | SRKW (J pod) | Stereotyped call S1 | 155.3±7.4 | Holt et al. [57] |
| San Juan Islands, WA | SRKW (J, K, and L pods) | Stereotyped calls | 155.1±6.5 | Holt et al. [54] |

The 300 Hz frequency bandwidth was selected, as we assume it is the smallest bandwidth necessary for an automated detector to detect killer whale pulsed calls. The frequency boundaries 1,000 and 14,800 Hz were chosen to cover the frequency range of killer whale pulsed calls [51,52].

Ambient noise level, NL, varies over time due to sounds from passing ships, weather, and breaking wave sound, and often from non-acoustic water flow noise over the sensor caused by tidal currents. The ambient sound levels at the hydrophones vary substantially over time due to these influences. To simulate the actual conditions at each model site, ambient sound levels were calculated from acoustic recordings collected by the shore-cabled DFO hydrophones and DFO AMAR recorders (Autonomous Multichannel Acoustic Recorders, JASCO Applied Sciences) or by JASCO AMARs deployed near DFO hydrophones if actual DFO data were unavailable.

The maximum detection range was calculated for each minute of the ambient sound data from the DFO or JASCO recordings; the probability of detecting a killer whale call at a given range was then taken to be the number of 1-minute recordings with a maximum detection range equal to or greater than that range divided by the total number of 1-minute recordings.

To estimate the maximum detection range of the killer whale pulsed calls, a Monte Carlo simulation was used to account for the measured variability in source levels and animal depths. Maximum detection ranges were calculated 10,000 times for all ambient sound levels available by randomly choosing 100 normally distributed source level values, with means and standard deviations from the literature (see below), and 100 animal depths from a log-logistic distribution based on animal tagging data (see below). Each iteration of the Monte Carlo process provided a probability of acoustic detection for each range. The distribution of the 10,000 detection probabilities obtained at the end of the Monte Carlo simulation are represented for each range by the percentiles 25, 50, and 75. The maximum detection range of SRKW discrete pulsed calls was estimated at eight PAM locations (Fig 1). The detection range of NRKW pulsed calls was estimated at three of those locations (Port Renfrew, Sheringham Point, and Swiftsure Bank). While the low-frequency components of killer whale pulsed calls are omnidirectional, the higher-frequency components exhibit some degree of directionality [28]. In most cases, the low-frequency components determine the maximum detection range, allowing us to estimate the true probability of detection. In some instances, however, the higher-frequency components set the maximum detection range. In these latter cases, because directionality is not incorporated into the Monte Carlo simulation, we can only estimate the probability of on-axis detection. To avoid any misuse of the results provided we will always refer to our results as probability of on-axis detection.

## Source level estimation

Pulsed calls were modeled in this analysis because they typically propagate better over greater distances than whistles and clicks and are used most often by the whales [26,53]. We used the source levels reported by Holt et al. [54] for SRKW (as they cover stereotyped calls from three different pods) and the source levels reported by Wladichuk et al. [55] for NRKW. The source levels for NRKW represent those of calls produced by two acoustic clans (A and G clans [51]). There are three clans in the NRKW population and clans show differences in habitat use. A and G clans have been reported [39,56] to use areas monitored by at least two of the PAM stations for which detection range was modeled (Swiftsure Bank and Port Renfrew-) and occasionally may travel further into Juan de Fuca Strait [56]. Source levels reported in the literature are described by their mean and standard deviation. To include the variability of the source level values in the detection range analysis, 100 source level values were randomly selected during the Monte-Carlo process from a Gaussian distribution defined by the mean and standard deviation from Holt et al. [54] and Wladichuk et al. [55].

The distribution of the energy of killer whale calls across frequencies is an important parameter that can affect how far a call can be detected. Unfortunately, all reported source levels in the literature are broadband (i.e., a single value for the entire frequency band of the killer whale calls). To address this, we analyzed 35 SRKW pulsed calls and 14 NRKW pulsed calls with high Signal-to-Noise Ratio (SNR). The SRKW call data set was collected with hydrophones deployed by the

DFO Cetacean Research Program, Ecosystem Science Division at various locations, and is composed of a selection of typical call types utilized by all SRKW pods, including types S1, S3, S4, S10, S13, S42, and S44 [58]. The NRKW calls were collected by a hydrophone deployed off northern Vancouver Island in the Johnstone Strait area in 2019 and provided by Dr. Jennifer Wladichuk (JASCO, University of Victoria). All calls/call types used in this study (S1 Fig) were annotated by bioacousticians specialized in killer whale acoustics with extensive experience in identifying call types and defining call type repertoires [13,51,58] using spectrogram and sound simultaneously. Calls are typically annotated as single calls and call bouts are used to identify specific resident killer whale pods, clans and populations. Killer whale presence is confirmed via photo identification of individuals and groups (matrilines and pods) [59]. The NRKW call source level estimates in Table 1 reflect those of a subset of call types used by NRKW predominantly by members of two acoustic clans (A and G clan), [51]) but could be calls from either clan, as both were present during the measurements. The measurements, however, may not represent the full range of source levels within call repertoires of both clans or the population as a whole. We assumed that the high SNR suggested that the vocalizing animals were very close to the hydrophone and the received sound levels reflect those of the direct path between calling whale and hydrophone, therefore these calls could be used to estimate the distribution of energy across frequencies at the source.

The PAMlab software (JASCO Applied Sciences), was used to annotate the killer whale calls (Fig 2). A custom MAT-LAB script was then used to calculate the sound pressure level (SPL) in each 300 Hz frequency band (Fig 2 c-d) to match the minimum bandwidth that is accepted by the automated detector used to analyze data from the PAM stations. The script next performed a linear fit to the measured data (least-square fit) between 1,000 and 14,800 Hz to model the slope of the SPL across frequencies (Fig 2, red line). The slope of the fitted line was −0.66 dB per 300 Hz band for SRKW and −0.27 dB per 300 Hz band for NRKW calls. Source levels used in the Monte-Carlo process were all represented with this frequency distribution.

## Vocalization depth

Propagation loss is depth dependent, so the depth at which an animal vocalizes can greatly impact how far from the hydrophone its call can be detected. This was accounted for in the Monte Carlo process by repeating the detection range calculations for 100 source depths randomly selected from a distribution representing the typical depth distribution of SRKW, modeled using *in situ* data collected by Wright et al. [60] from 34 archival Dtags [61] attached to 32 individual NRKWs located between Northern Vancouver Island and mainland British Columbia.

A log-logistic model, $f(z|\mu,\sigma)$, was fitted to the tag data using Maximum Likelihood:

$$f(z|\mu,\sigma) = \frac{e^y}{\sigma z (1 + e^y)^2},$$

(6)

where $y = \frac{\log(z)-\mu}{\sigma}$, $z$ is the animal's depth ($z \geq 0$), $\mu = 2.0212$, and $\sigma = 0.7739$. Because the hydrophones are located in water depths less than 200 m, the log-logistic model was created using all the depths from the tag data less than 200 m. Fig 3 shows the probability distribution of the empirical data collected by the tag (n = 2,215,700) and the probability distribution of 2,215,700 depth samples randomly drawn from the log-logistic model.

## Propagation loss modeling

Detection ranges were modeled at each of the DFO PAM station sites (Table 2 and Fig 4). For each location, propagation loss was modeled along transects in different directions (Fig 4, red lines. S1 Table), to sample the propagation loss characteristics at multiple depths as a function of range and azimuth from each location. JASCO's Marine Operations Noise Model (MONM) was used to calculate the propagation loss. MONM takes as inputs the source location and depth, a geoacoustic profile based on the sediment properties, a sound speed profile for the water column, and a record of the

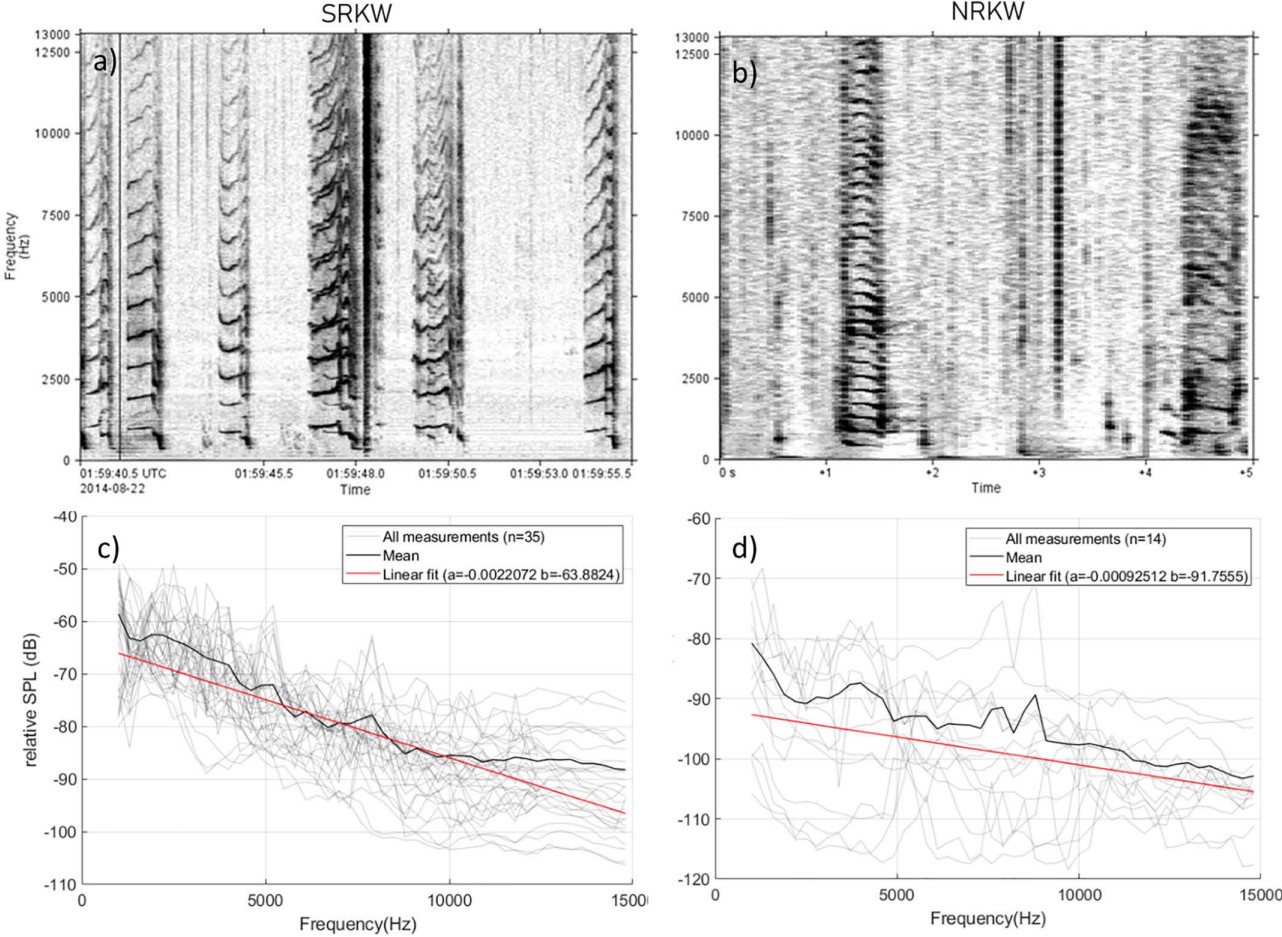

**Fig 2. Frequency distribution of killer whale calls.** Spectrogram of a selection of (a) SRKW pulsed call recordings and (b) NRKW pulsed call recordings used to determine the energy distribution of killer whale calls across frequency. Relative spectral distribution of (c) SRKW call source levels and (d) NRKW call source levels in 300 Hz bands across 1-15 kHz range based on 35 and 14 high SNR calls, respectively, recorded from small vessels.

bathymetry along the modeled transects. MONM computes range-, depth-, and frequency-dependent propagation loss using either a split-step Padé solution [62] to the wave equation (used here for frequencies below 2.5 kHz) or a ray-tracing engine based on the Bellhop algorithm [63], which was used here for frequencies above 2.5 kHz. The model accounts for absorption, which can be important at the frequencies considered in this study. Calculations were computed at three frequencies within each 300 Hz band and averaged to provide a propagation loss estimate for each frequency band.

Five arc second ocean bathymetry data for the Salish Sea area were obtained from the high-resolution NOAA digital elevation model [64]. These data were supplemented to the north and west with lower resolution soundings from a Canadian Hydrographic Service data set provided, with permission, by Nautical Data International Inc. Bathymetry data were re-projected onto a UTM Zone 10N coordinate grid with 100 × 100 m resolution, for use with MONM.

Sound propagation in shallow water is strongly influenced by the geoacoustic properties of the seafloor. These include the density, the compressional wave (P-wave) speed, the shear wave (S wave) speed, the compressional wave attenuation, and the shear wave attenuation of the seabed sediments. MONM incorporates these parameters when calculating propagation loss. Geoacoustic profiles for each model site were constructed to simulate the major features of the

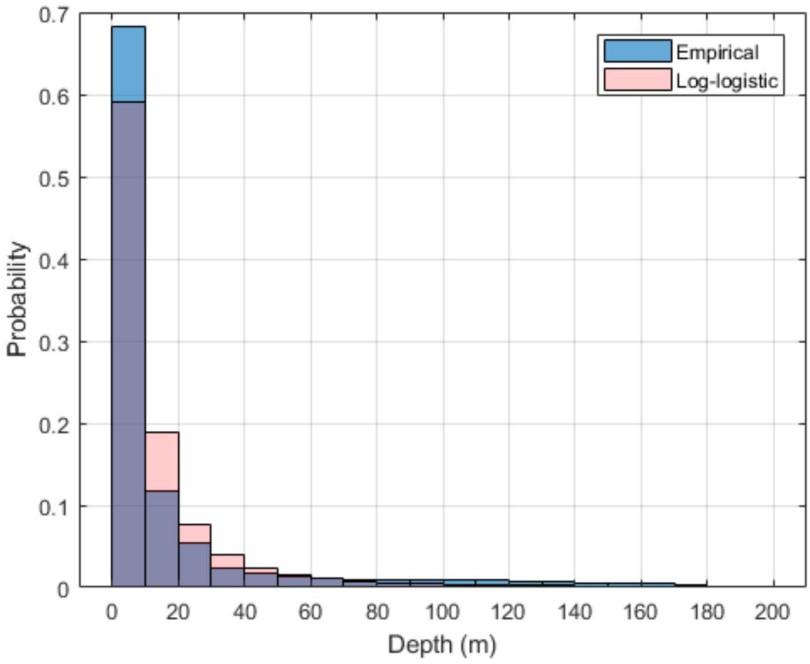

**Fig 3. Dive depth probability distribution and distribution of samples drawn from the log-logistic model for the Monte Carlo simulation.**

**Table 2. Model site coordinates.**

| Model site | Latitude (° N) | Longitude (° W) | Easting | Northing | Water Depth (m) |
|---|---|---|---|---|---|
| East Point | 48.781 | −123.049 | 496427.0 | 5403087.9 | 16 |
| Enterprise Reef | 48.847 | −123.348 | 474502.9 | 5410499.6 | 19 |
| Mouat Point | 48.777 | −123.319 | 476546.4 | 5402738.1 | 16 |
| Port Renfrew | 48.504 | −124.517 | 387952.7 | 5373481.3 | 172 |
| Sheringham Point | 48.376 | −123.921 | 431764.4 | 5358486.1 | 15 |
| Sturdies Bay | 48.876 | −123.309 | 477306.3 | 5413706.9 | 12 |
| Swiftsure Bank | 48.515 | −124.936 | 357034.3 | 5375393.4 | 75 |
| Tilly Point | 48.732 | −123.205 | 484910.2 | 5397719.7 | 17 |

sediments. The geoacoustic properties for all the model sites were estimated based on the mean grain size from surface and sub-surface sediment grain-size samples from the Geological Survey of Canada (GSC) and the BC Marine Ecological Classification (BCMEC) maps [65]. Geoacoustic parameters were then computed based on Buckingham's grain-shearing model [66]. The grain-shearing model computes geoacoustic properties of seabed sediment (sand, silt, clay) from porosity and grain size. Four geoacoustic profiles were created based on location and the geological data (S2 Fig, S2 Table). The colored bubbles in Fig 4 indicate which profile (A through D) was used at each model site.

Water column sound speed profiles for summer and winter were derived from data collected by DFO in 2018 and 2019 near each of the model sites. These were augmented, typically with deeper depth data, from the US Naval Oceanographic Office's Generalized Digital Environmental Model (GDEM) V 3.0 [67,68]. GDEM provides an ocean climatology of temperature and salinity for the world's oceans on a latitude-longitude grid with 0.25° resolution, with a temporal resolution of one month, based on global historical observations from the US Navy's Master Oceanographic Observational Data Set (MOODS). At each site, the CTD data were first extended to the deepest water depths (i.e., 400 m except 250 m for

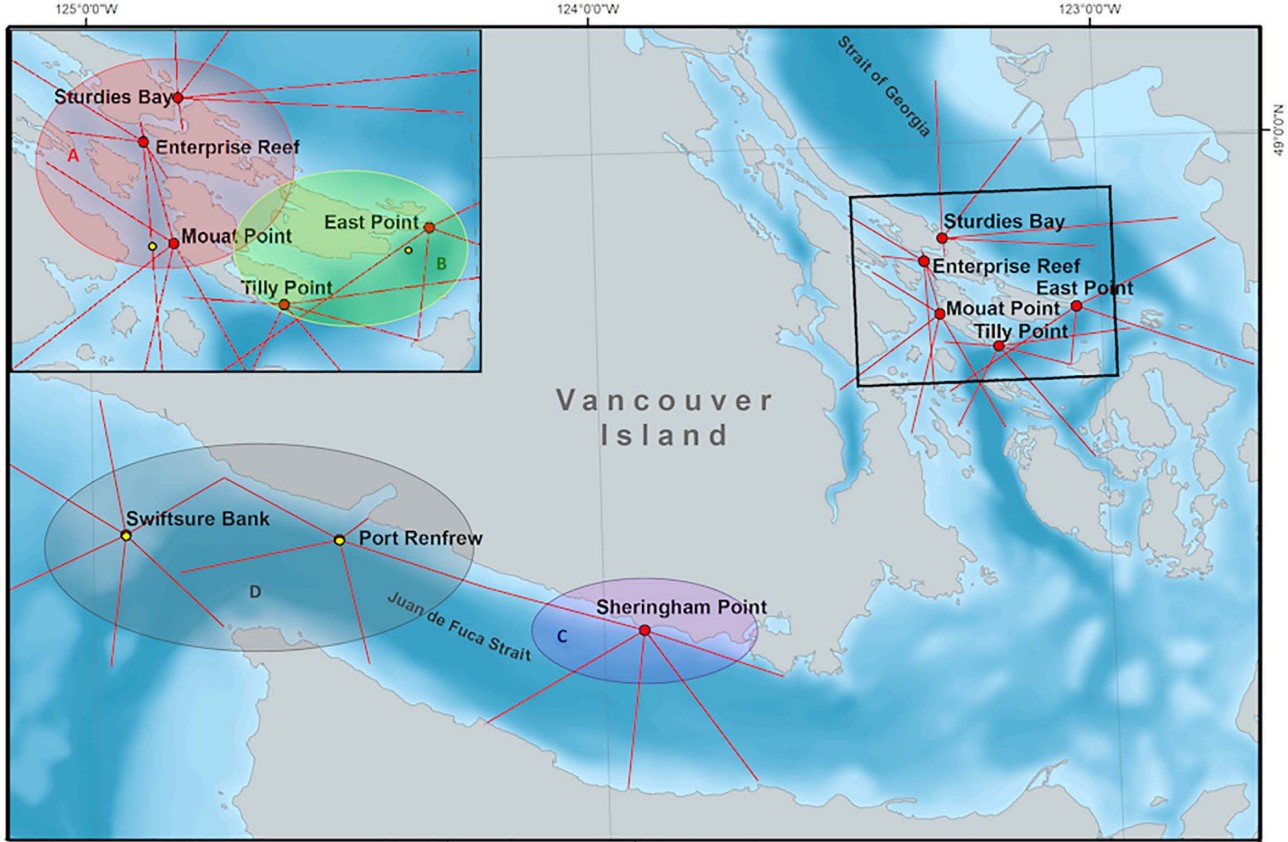

**Fig 4. Map of the model sites with transects used for propagation loss computations.** Color-shaded bubbles group sites according to geoacoustic properties. Yellow circles denote locations of ambient sound measurements. Red lines correspond to the transects.

Sheringham Point). The temperature-salinity profiles were converted to sound speed profiles according to Coppens [69]. The data were then smoothed using boxcar averaging and interpolated to get values at specified depths for modeling. If applicable, multiple profiles were averaged to compute the final profile. Winter and summer sound propagation conditions are very distinct (S3 Fig). In winter, cooler surface waters, wind-driven mixing, and atmospheric cooling result in a strong surface-duct sound profile. In summer, heating of the surface results in a downward refracting sound profile at most of the sites. Note that we did not use a full wave acoustic propagation model like in Helble et al, 2022 [45] but used MONM to model propagation losses on a wide frequency range, which in our case was more computationally efficient and scalable to multiple locations.

### Ambient sound level characterization

Ambient sound levels were calculated from acoustic recordings collected by the DFO-PAM hydrophones or by Autonomous Multi-Channel Acoustic Recorders (AMARs; JASCO) deployed near PAM sites when the DFO data were unavailable (Fig 4 yellow circles; S3 Table). Ambient sound levels were obtained from approximately one month of acoustic data collected at each site and season (winter vs summer). Table S3 indicates the time periods selected for each season, as well as the coordinates and depths of the acoustic recorders used. The East Point winter data were collected by an AMAR deployed by JASCO and were used with permission from Vancouver Fraser Port Authority and Transport Canada. The

acoustic data used for the Mouat Point site (summer) were collected by an AMAR and were provided for this analysis by JASCO Applied Sciences. Acoustic recordings at the Port Renfrew and Swiftsure Bank site were collected by AMARs and were provided by Dr. Svein Vagle (DFO). All other acoustic recordings were collected by the cabled icListenHF hydrophones (OceanSonics) deployed by DFO. All acoustic data acquired with AMARs used an M36-V35-100 calibrated omnidirectional hydrophone (GeoSpectrum Technologies Inc.; −165±5 dB re 1 V/μPa nominal sensitivity) and a gain of 0 dB.

Ambient sound processing was performed using the PAMlab software (JASCO Applied Sciences). The raw pressure waveform data were scaled according to the mean calibrated pressure sensitivity of the recorder and adjusted for the frequency response of the hydrophone sensor. Time domain pressure waveforms were analyzed to find SPL for each minute of data. SPL, which is a physical measure of sound amplitude, can be averaged over time and integrated over different frequency bands. For this study, SPL was calculated for each 300 Hz frequency band between 1000 and 14800 Hz (46 frequency bands in total). The frequency boundaries 1000 and 14800 Hz were chosen to cover the frequencies of killer whale pulsed calls [51,52].

## Results

The 100 randomly drawn broadband source levels from the Monte Carlo simulations for each model site approximate the normal distribution from which they were drawn (S4 Fig, S5 Fig). The broadband source level samples ranged in value from 135 to 175 dB re 1 μPa, and 120–170 dB re 1 μPa for SRKW and NRKW, respectively. Simulated source levels were maximum at low frequencies (S6 Fig and S7 Fig), as was the model used for the distribution that the band levels were drawn from. Distributions of vocalization depths did not exceed 100 m depth (S8 Fig and S9 Fig) and most of the simulated calls depths were within the top 20 m of the water column. Background sound levels at each model site were highest at 1000 Hz and decreased as frequency increased (S10 Fig and S11 Fig). The mean band levels were all below 100 dB re 1 μPa. At several sites, sound levels were higher in winter than in summer likely attributable to weather. The propagation loss modeling results show differences across sites under the influence of the local bathymetric variation. A winter surface sound duct supports sound propagation in the top 20 m of the water column at the shallower model sites and is notable for Sturdies Bay. In general, propagation loss is greater in the summer compared to the winter at East Point, Mouat Point, Sheringham Point, and Sturdies Bay while lower in the summer than in winter at Port Renfrew. There is little seasonal dependence of the propagation loss at Enterprise Reef, Swiftsure Bank, and Tilly Point. Fig 5 shows an example of propagation loss modeled at Sturdies Bay in the 1900–2200 Hz band for winter and summer conditions. S14 Fig and S15 Fig show examples propagation loss modeled at the other locations.

Table 3 summarizes the median detection range for all sites, call types, and seasons. Contours depicting the median detection range at each site for P=0.5 for SRKW in summer and winter are shown in Fig 6. All detection range probabilities are plotted in Fig 7 and Fig 8 for SRKW and NRKW, respectively. Detection range estimates suggest that the East Point, Sturdies Bay, Tilly Point, and Enterprise Reef listening stations have the longest median detection ranges for SRKW pulsed calls (50% probability of on-axis detection at 2.2, 4.2, 2.4, and 2.0 km, respectively, in winter; 7.9, 2.0, 2.0, and 2.2 km, respectively, in summer). The shortest SRKW median detection range (50% probability of on-axis detection) is modeled to be 650 m in winter, at Swiftsure Bank; in summer it is 1.1 km at Mouat Point. There is a 10% probability of on-axis SRKW pulsed calls being detected at median ranges between 1.5 and 25 km in winter or between 4.2 and 37.5 km in summer (site dependent). Pulsed calls can be detected with a 90% probability of on-axis detection at site-dependent, median ranges between 130 and 540 m in winter and between 90 and 800 m in summer. The set of NRKW pulsed calls included in the study have smaller detection ranges than SRKW pulsed calls for all sites and seasons. The longest median detection ranges for NRKW pulsed calls are reached at Sheringham Point with 750 m in winter and 600 m in summer for P=0.5. Swiftsure Bank has the shortest median detection range distance for NRKW pulsed calls; 240 m in winter and 510 m in summer (P=0.5). For most sites, the detection ranges are shorter during winter than summer with the exception of Sturdies Bay for SRKW (Fig 7f) and Sheringham Point for NRKW (Fig 8b)

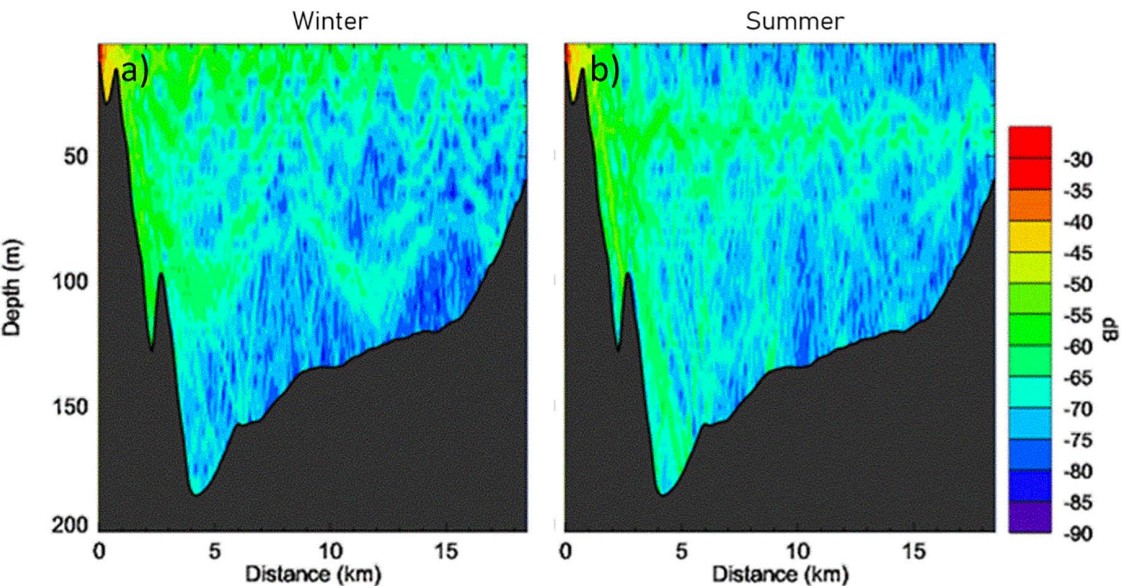

**Fig 5. Modeled propagation loss at Sturdies Bay in a) winter and b) summer for the 1900–2200 Hz band, as a function of range and depth along a single radial at a bearing of 40° (across the Strait of Georgia).**

**Table 3. Median detection range (in meters) for probability of on-axis detection (P) of 0.1, 0.5, and 0.9.**

| Model Site | Call type | Season | P = 0.1 | P = 0.5 | P = 0.9 |
|---|---|---|---|---|---|
| East Point | SRKW | Winter | 21,000 | 2,200 | 340 |
| | | Summer | 21,700 | 7,900 | 800 |
| Enterprise Reef | SRKW | Winter | 25,000 | 2,000 | 150 |
| | | Summer | 21,600 | 2,200 | 160 |
| Mouat Point | SRKW | Winter | 12,900 | 1,400 | 150 |
| | | Summer | 37,500 | 1,100 | 90 |
| Port Renfrew | SRKW | Winter | 2,600 | 1,000 | 400 |
| | | Summer | 4,500 | 1,400 | 470 |
| | NRKW | Winter | 900 | 300 | 160 |
| | | Summer | 2,000 | 850 | 400 |
| Sheringham Point | SRKW | Winter | 1,500 | 850 | 440 |
| | | Summer | 4,200 | 1,600 | 470 |
| | NRKW | Winter | 1,400 | 750 | 420 |
| | | Summer | 1,900 | 600 | 210 |
| Sturdies Bay | SRKW | Winter | 10,800 | 4,200 | 130 |
| | | Summer | 5,300 | 2,000 | 130 |
| Swiftsure Bank | SRKW | Winter | 2,000 | 650 | 270 |
| | | Summer | 6,500 | 1,600 | 380 |
| | NRKW | Winter | 680 | 240 | 150 |
| | | Summer | 1,600 | 510 | 190 |
| Tilly Point | SRKW | Winter | 8,600 | 2,400 | 540 |
| | | Summer | 8,600 | 2,000 | 480 |

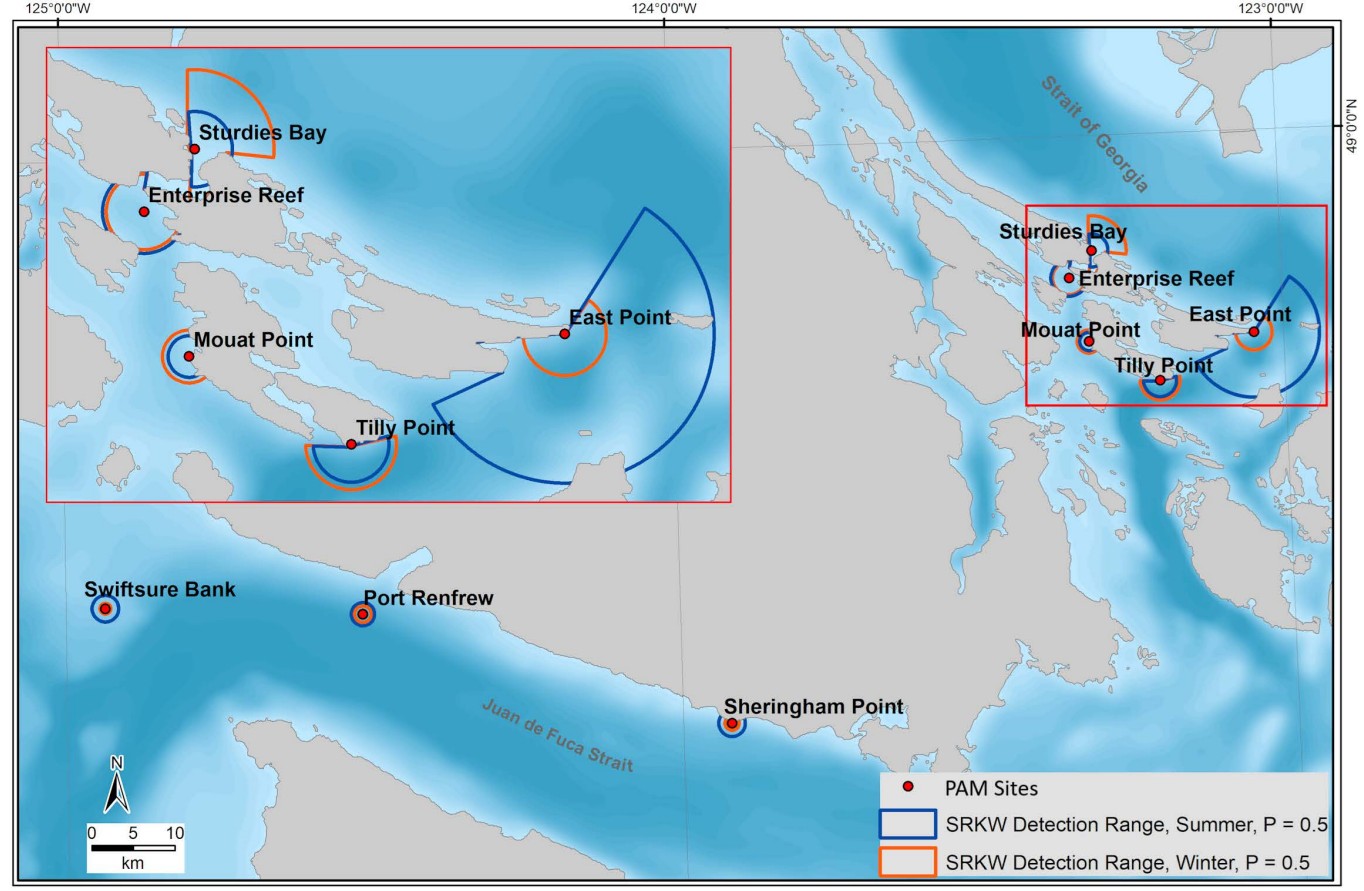

**Fig 6. Median detection ranges for probability of on-axis detection, P, of 0.5 for SRKW in summer (blue) and winter (orange) at each model site.**

The maximum detection ranges for SRKW calls were most often associated with the lower frequency bands (below 8 kHz) but varied at each site and season due to the various ambient sound and propagation loss conditions (Fig 9, S12 Fig and S13 Fig). The maximum detection ranges for the set of NRKW calls were most often associated with higher frequency bands (above 10 kHz).

A sensitivity analysis was conducted to better understand which factors most influence the acoustic detection range results. This analysis focused on the acoustic detection range results for NRKW at Port Renfrew because NRKW have been observed at these locations but not at the other modeling sites. Detection ranges were recalculated several times by modifying in turn each of the following factors of the Monte Carlo simulation: source level frequency distribution (uniform versus negative slope, S16 Fig), ambient sound levels (winter versus summer, S17 Fig), the propagation loss (winter versus summer, S14 Fig, S15 Fig), and the animal depth distribution (uniform versus loglogistic, S18 Fig). Table 4 summarizes the results of the sensitivity analysis; the resulting detection range probabilities are compared in Fig 10. Fig 11 shows the frequency bands responsible for the maximum detection ranges for each of the two considered source level frequency distributions. This analysis showed that the frequency distribution of the source levels had the strongest effect on the detection range probability, the seasonal ambient sound levels had the second strongest influence (maximum detection ranges were longer in summer), then the animal depth distribution (maximum detection ranges were longer when whales were assumed to spend most of their time near the surface), and finally the seasonal propagation loss had a minimal effect on the detection range probabilities.

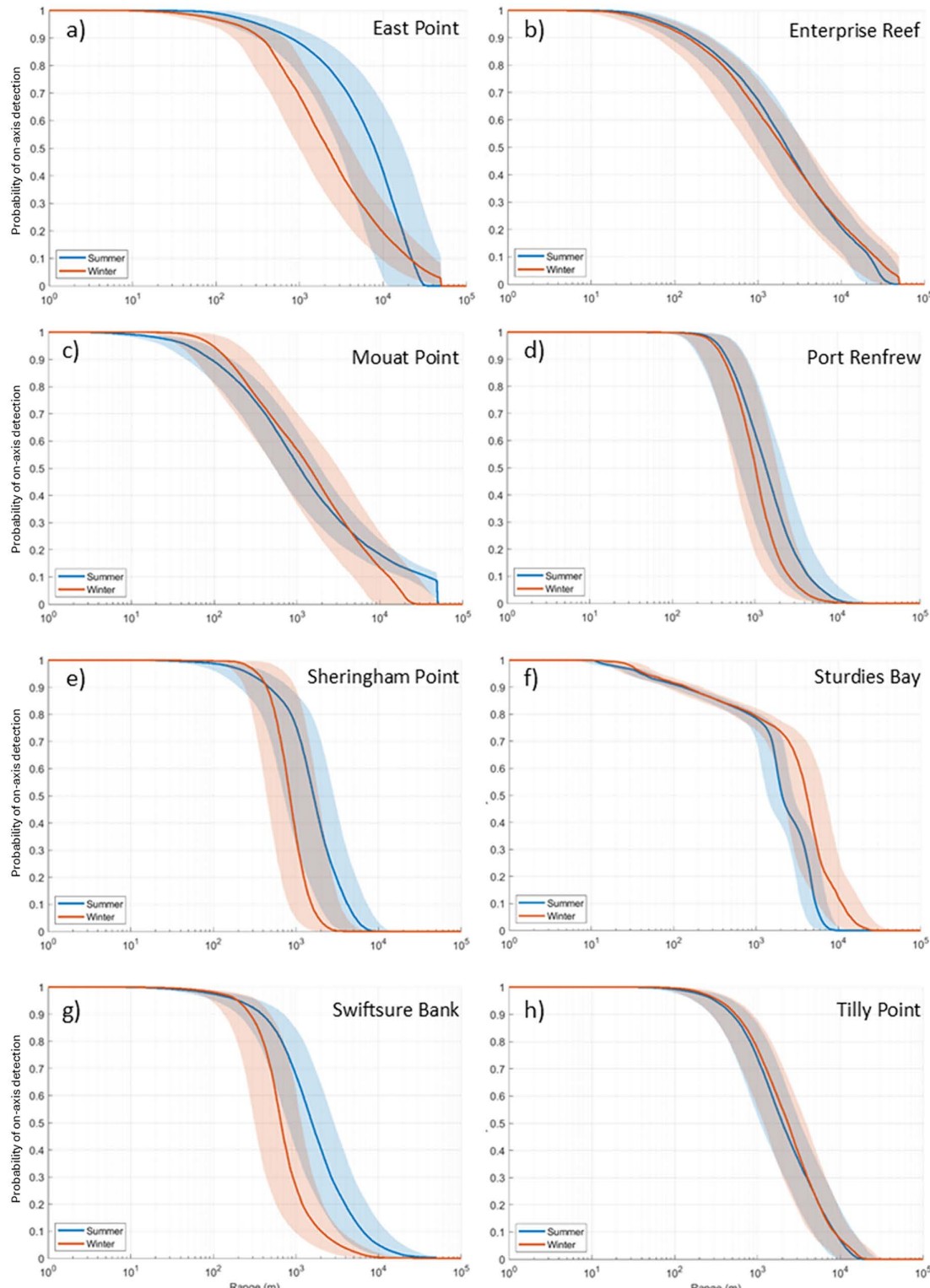

**Fig 7. Probabilities of on-axis detection for SRKW pulsed calls in summer (blue) and winter (orange).** The solid lines are the median values, and the shaded areas define the 25th and 75th percentiles. Results are shown for (a) East Point, (b) Enterprise Reef, (c) Mouat Point, (d) Port Renfrew, (e) Sheringham Point, (f) Sturdies Bay, (g) Swiftsure Bank, and (h) Tilly Point.

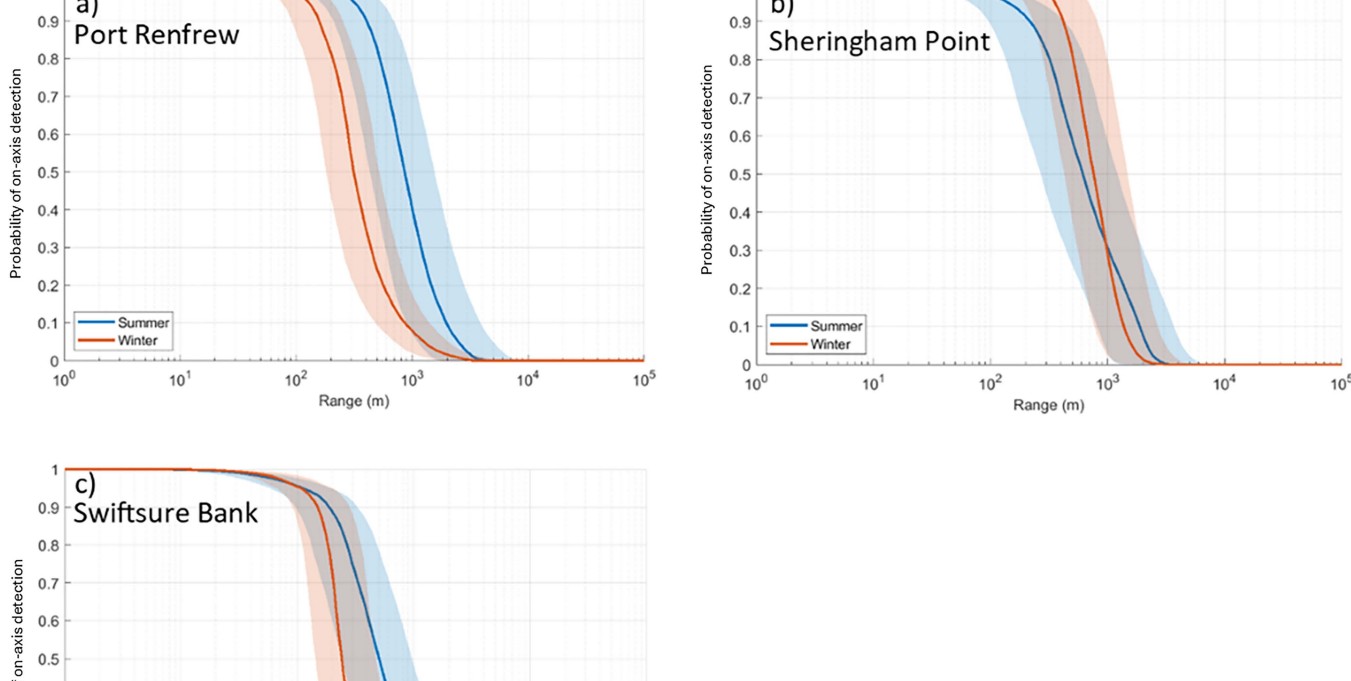

**Fig 8. Probabilities of on-axis detection for NRKW pulsed calls in summer (blue) and winter (orange).** The solid lines are the median values, and the shaded areas define the 25th and 75th percentiles. Results are shown for (a) Port Renfrew, (b) Sheringham Point, and (c) Swiftsure Bank.

In order to better illustrate the effect of ambient noise levels on the acoustic detection range, the probability of on-axis detection of SRKW at East Point during the summer (Fig 7a, blue curve) was recomputed separately for high, medium and low ambient noise conditions (Fig 12). High conditions correspond to broadband levels > 94.06 dB re 1 μPa, medium to broadband levels between 87.47 and 94.06 dB re 1 μPa, and low to broadband levels < 87.47 dB re 1 μPa. SPL intervals for each condition were defined by the 33rd and 66th percentile of broadband values in order to have each condition represented equally (14,348 1-minute recordings). Fig 12 shows that for a same location and season the fluctuation of ambient noise levels substantially impact the acoustic detection range going from a median detection range of 1,640 m in the worst ambient noise conditions (high ambient noise conditions) to 15,500 m in the best conditions (low ambient noise conditions) for P = 0.5.

## Discussion

Many studies estimating the acoustic detection ranges of baleen whale calls, such blue, fin and right whales use source level, background noise levels, and propagation losses estimated over a very narrow frequency band [44,48]. Detection range estimates become more challenging when examining broadband sounds such as killer whale pulse calls. Miller [27] has estimated the detection range of resident killer whales in British Columbia by conducting acoustic localization

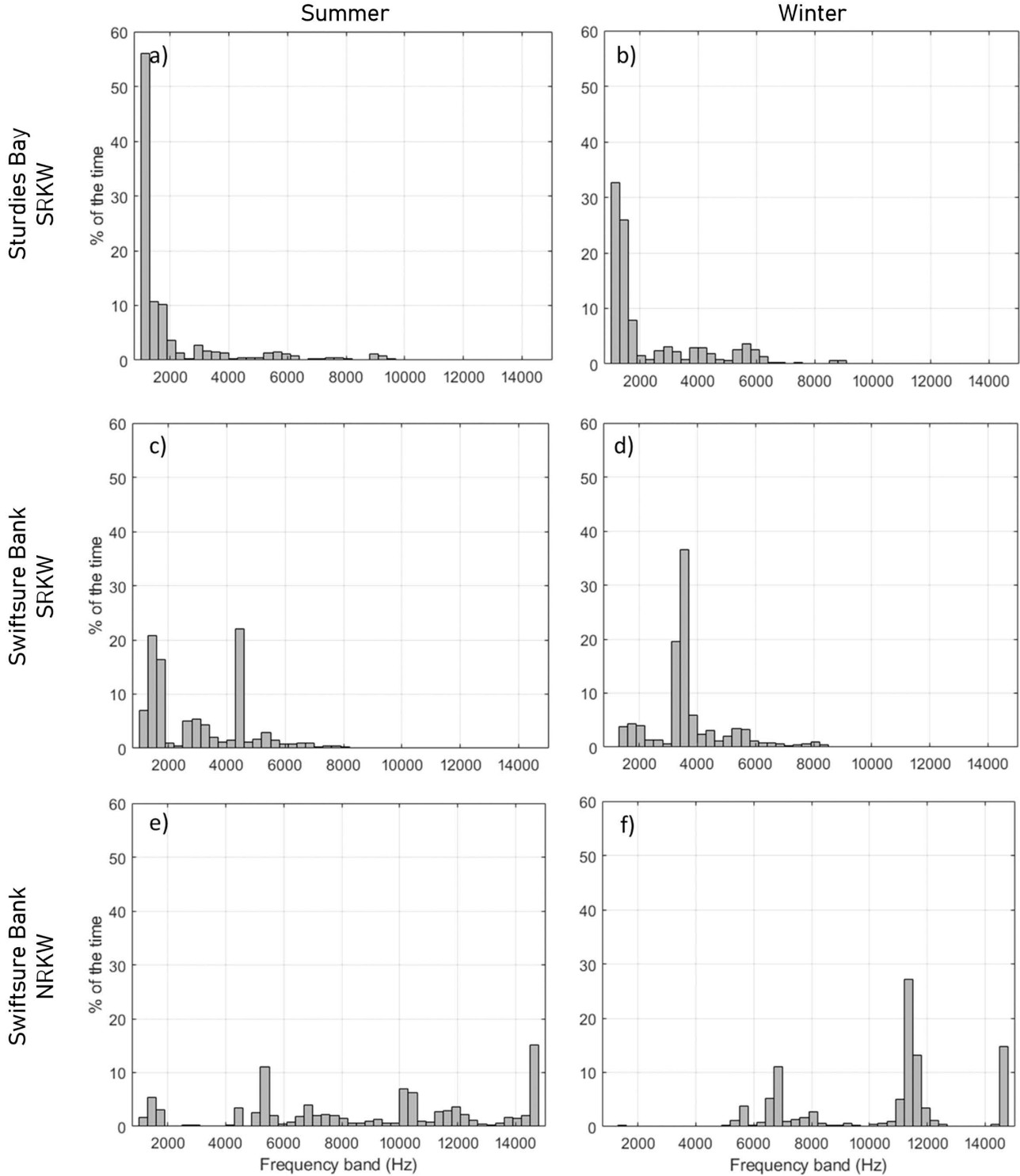

**Fig 9. Percentage of the results for which each frequency band yielded the maximum acoustic detection range from the Monte Carlo simulation.** Results are shown for SRKW at Sturdies Bay in (a) summer and (b) winter, for SRKW at Swiftsure Bank in (c) summer and (d) winter, and for NRKW at Swiftsure Bank in (e) summer and (f) winter. Results for all other locations are in S12 Fig and S13 Fig.

**Table 4. Sensitivity analysis based on calculations for Northern Resident Killer Whale calls (NRKW-A) at Port Renfrew. Difference of the median detection range for P = 0.5 for each of the Monte Carlo factors tested.**

| Factor | Median detection range difference for P = 0.5 (m) |
|---|---|
| Frequency distribution of source levels (no slope – negative slope) | 850 (1,700–850) |
| Ambient sound levels (summer-winter) | 600 (850–250) |
| Vocalization depth distribution (log-logistic – uniform) | 230 (850–620) |
| Propagation loss (winter – summer) | 30 (880–850) |

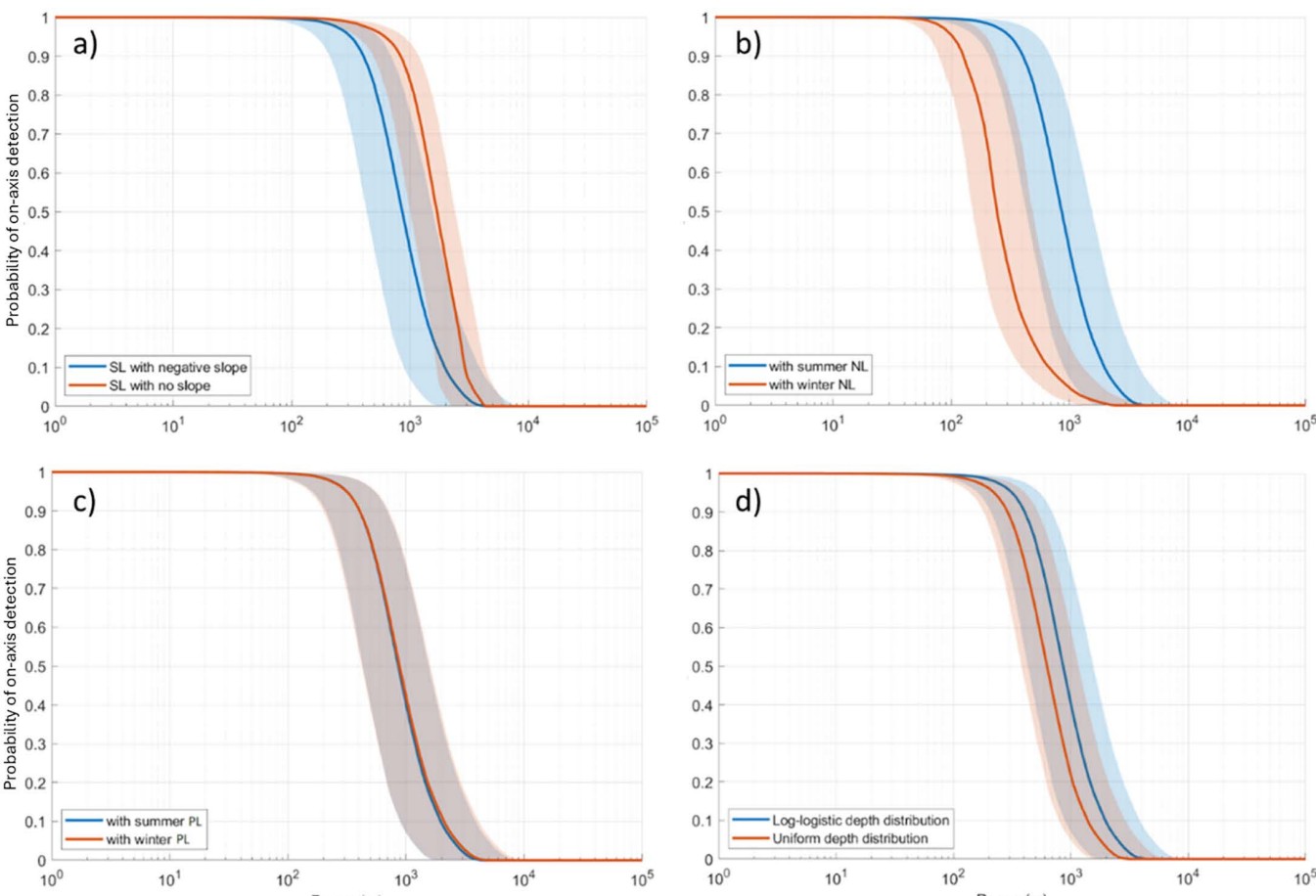

**Fig 10. Sensitivity analysis: Effect of changing the source level frequency distribution on the detection range, the seasonal ambient sound levels, the vocalization depth distribution, and the seasonal propagation loss.** Detection range probabilities for NRKW pulsed calls at Port Renfrew comparing in turn (a) a negatively sloped frequency distribution of source levels (blue) and a uniform frequency distribution of source levels (orange) in summer, (b) summer ambient sound levels (blue) and winter ambient sound levels (orange), (c) summer propagation losses (blue) and winter propagation losses (orange), and (d) a log-logistic animal depth distribution (blue) and a uniform animal depth distribution (orange) for summer. In each panel the solid lines are the median values and the shaded areas define the 25th and 75th percentiles.

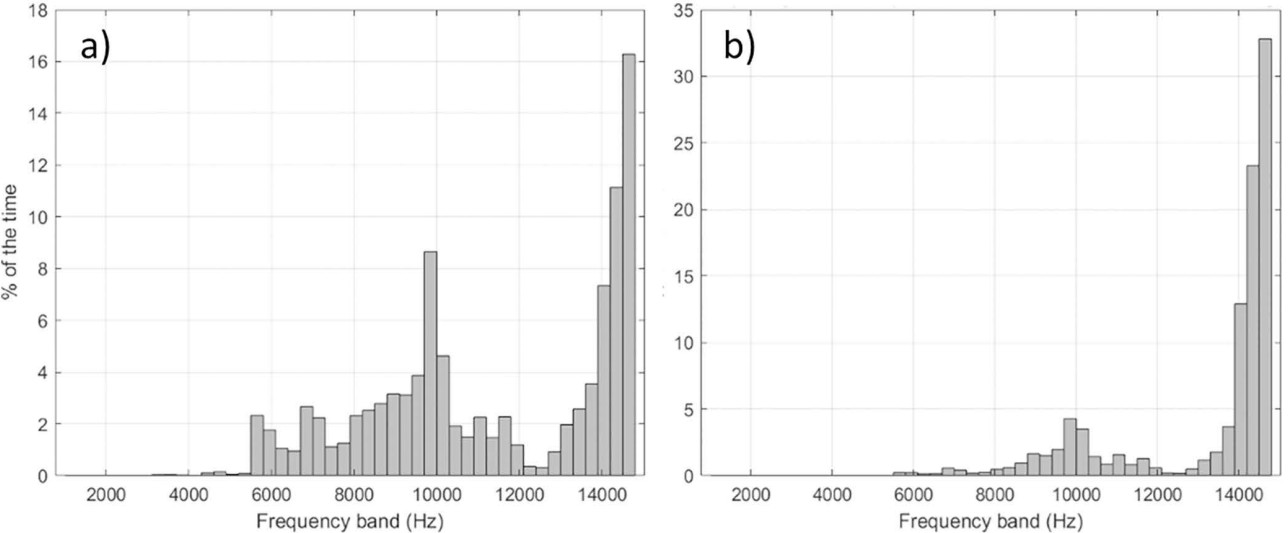

**Fig 11. Sensitivity analysis: Effect of the source level frequency distribution on the detection bands on the frequency of detection.** Percentage of the results for which each frequency band yielded the maximum detection range from the Monte Carlo simulation for NRKW in summer, using (a) a negatively sloped frequency distribution of source levels and (b) a uniform frequency distribution of source levels.

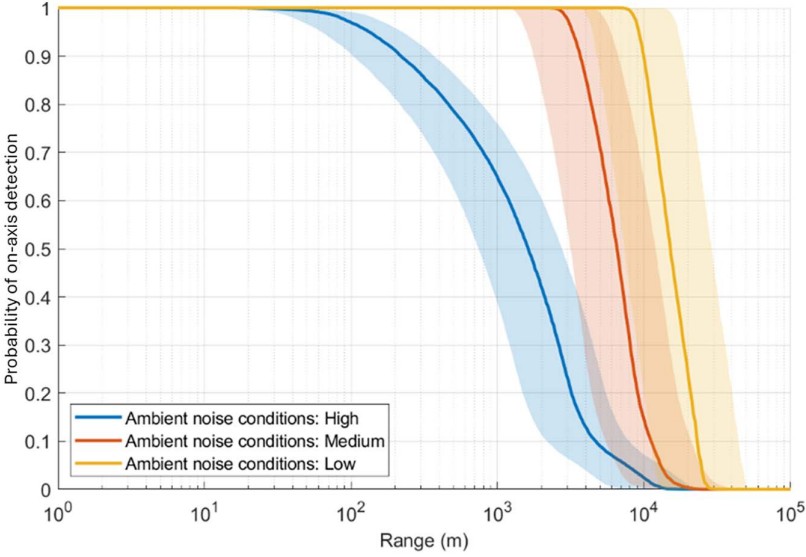

**Fig 12. Probabilities of on-axis detection for SRKW pulsed calls in summer at East Point calculated separately for high (blue), medium (red), and low (orange) ambient noise conditions.** The solid lines are the median values, and the shaded areas define the 25th and 75th percentiles.

to measure source level in third-octave bands and using simplified propagation loss equation. While this study provided insights on killer whale communication range, it did not take into account how the animal vocalization depth and seasonal water column properties (e.g., changes in sound speed profiles) affected the propagation loss and approximated noise condition based on sea-state rather than in-situ measurements. Such effects are important to understand and account for, particularly in the context of noise mitigation. Our approach, based on Monte Carlo simulations [70], took into account the

variability of the propagation losses with seasons, animal depths and frequency, the variability of the interfering background noise, and the frequency dependent variability in source levels. Other studies based on Monte Carlo simulations and acoustic propagation modeling have estimated detection probabilities for beaked, humpback whale and delphinid sounds [45,71–73]. Modeling studies such as [72] have shown good agreement between *in situ* detection results and model predictions, which suggests that simulation studies can provide reliable detection range estimates. Comparing our simulation results with field measurements is a goal for future efforts.

In this study we evaluated the detection range of SRKW and NRKW calls at different seasons and different locations in the Salish Sea. Detection ranges fluctuated substantially between locations. Most PAM stations around the Southern Gulf Islands (Sturdies Bay, Enterprise Reef, Tilly Point and East Point) had larger detection ranges (i.e., median detection ranges for P=0.5, Fig. 6) than stations located in the Strait of Juan de Fuca (Sheringham Point, Port Renfrew) and on Swiftsure Bank. This difference is driven by the higher ambient sound levels measured in the Juan de Fuca Strait than in the protected inshore waters of the Salish Sea that are favoring acoustic masking (S10 Fig and S11 Fig). This difference in ambient sound levels observed from our data is consistent with other acoustic measurements conducted in the Salish Sea [17]. The Juan de Fuca Strait is a major shipping lane funneling commercial vessels transiting from the Pacific to several ports in both the USA and Canada, including the Port of Vancouver, which is the largest port in Canada and the third largest in North America by tons of cargo [74]. Because it is facing the Pacific Ocean, the soundscape in the Juan de Fuca Strait is influenced by both local and offshore sources. This makes this area sometimes naturally noisier than the inshore waters of the Salish Sea, which increases acoustic masking of killer whale calls and reduces the detection range. Mouat Point is one of the locations with short median detection ranges. This PAM station is located at the intersection of several ferry routes connecting Vancouver Island, the Southern Gulf Islands, and Vancouver on Canada's mainland. As a result, it is noisier than the other locations in the area (S10 Fig), leading to lower detections ranges. In winter, Sturdies Bay has lower ambient sound levels than the other locations (S10 Fig) which results in greater detection ranges (Fig 7). Sturdies Bay is in Active Pass, which is a major thoroughfare for both commercial vessels including ferries and fishing vessels, as well as recreational vessels. The latter are more frequently present during the summer when there are also many more ferries transiting through the Pass compared to winter [75]. This is supported by a lower number of acoustic whale detections in Active Pass in summer versus winter compared to detections made by onshore sighters and thermal imaging cameras (Yurk, H. Pers. Comm.). While this location may indeed be quieter, ambient sound data from this location and season were more limited than others (n=5126), and ambient sound conditions may be underestimated. In summer, East Point has the largest detection range for SRKW. This is mostly due to the lower ambient sound levels in the frequency band 2–3 kHz (Fig S10) which is the band responsible for more than 50% of the maximum detection range realizations (Fig S12). The East Point PAM is in Boundary Pass, which is a waterway used by commercial vessels on their way to and from the Port of Vancouver but is not used by ferries. While ferries travel on regular schedules throughout the day, freight carrying vessels pass the East Point PAM on a more intermittent schedule sometimes with long periods between vessels. This may have influenced the ambient sound level measurements. Furthermore, in contrast to the Tilly Point PAM, which is also located in Boundary Pass but its soundscape is also influenced by activities in two adjacent straits. The East Point PAM is a little bit sheltered from sounds coming from other areas than Boundary Pass.

SRKW median detection ranges (P=0.5) are longer in summer than winter at five of the eight modeled locations (Table 3) and similar at two locations (Mouat Point and Tilly Point). The difference between summer and winter is mainly due to the seasonal variability in ambient sound levels at these locations. Ambient sound in the Salish Sea, as in most places, is known to be consistently more elevated in the winter than in the summer due to stronger winds [17,76]. Sturdies Bay is an exception and has shorter detection ranges in summer than winter. This is most likely due to the under sampling of ambient sound conditions in winter due to the limited data available and due to the aforementioned differences of the ambient conditions in summer and winter. Several types of acoustic recorders were used to collect the ambient data used for the summer and winter modeling. Differences in the mooring configurations and recorder depths could result in differences between the summer and winter ambient conditions that are not strictly due to natural seasonal differences.

Detection ranges for the subset of calls of NRKW that were investigated are shorter than those of SRKW (Table 3). This difference is explained by the different source levels of these NRKW calls versus SRKW calls. Stereotyped pulsed calls were from either one or two acoustic clans of NRKW collected in a quieter environment than the locations of the modeled detection range. The NRKW calls have a reported mean source level approximately 7 dB lower than SRKW (Table 1) and this is consequently the reason why NRKW calls were masked at much shorter distances. The frequency components of the killer whale calls responsible for the maximum detection range differ between NRKW and SRKW. Maximum detection ranges tend to be mostly achieved from the higher frequency component of the calls of NRKW (i.e., >6 kHz, Fig S13) while maximum detection ranges for SRKW calls were largely reached at lower frequencies (i.e., <4 kHz, Fig S12). However, some of the NRKW calls showed higher source levels at frequencies above 5 kHz than at lower frequencies (Fig 2d) and localized level maxima were also present in higher frequencies in SRKW calls (Fig 2 b). This is likely the result of calls with more than one distinct frequency component, a low frequency (LF) and a high frequency (HF) component (Fig 13, [77]). The NRKW calls were recorded with a hydrophone array that was dynamically placed in close proximity to the whales, which may have increased the influence of the higher directionality of HF components on the source level estimation. It should also be noted that the sample size of the NRKW calls is relatively small.

The HF component transmits more directionally [28] and therefore shows a higher source level than the corresponding frequency of the LF component when transmitted over shorter distances. While directionality likely accounted for some of the variability of the broadband source levels, we did not have enough information about the directivity patterns of the collected calls by frequency to be able to incorporate it in the detection range estimates. Directional HF call components, however, attenuate over shorter distances than LF components. Maximum detection range may therefore be primarily influenced by the range of omnidirectional LF call components. Call source levels may not be static and may vary based on the ambient sound level in which they are produced. Holt et al. [57] suggested that SRKW increase broadband call source levels in noisier environments (i.e., Lombard effect). This effect was not considered in the present study, as the

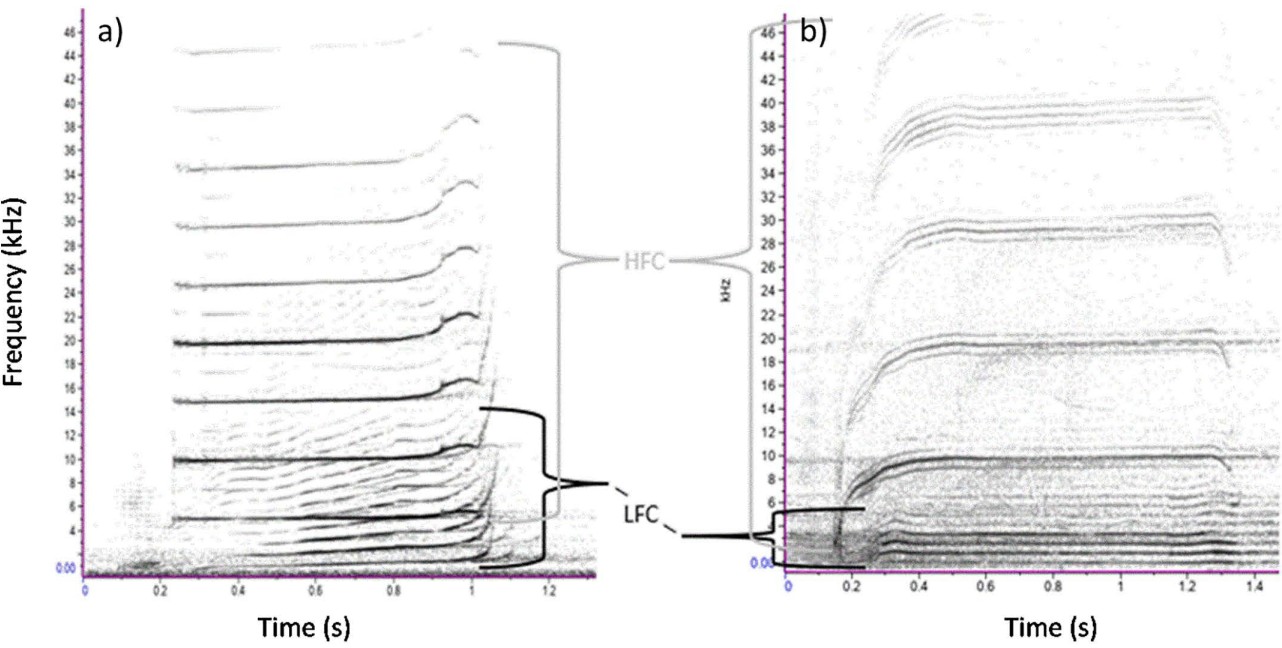

**Fig 13. Examples of two-component discrete pulsed calls of a) SRKW and b) NRKW.** The lower frequency components (LFCs) are transmitted omnidirectionally while the higher frequency components (HFCs) that may be produced parallel to the LFCs transmit more directionally (Miller 2002).

Lombard response reported by Holt et al. was based solely on broadband source levels. In contrast, detection ranges in this study were estimated using variation in source levels across the 500 Hz to 15 kHz band, in 300 Hz steps. No information is currently available regarding Lombard responses at this finer 300 Hz band resolution, and assuming an evenly distributed Lombard response across the modeled frequency range is unlikely, given the sound production in odontocetes may occur in registers [78]. Incorporating this effect to the detection range modeling is possible but would require a more comprehensive characterization (e.g., more call samples from multiple locations and a better understanding of the frequency dependence of the response) to benefit the detection range estimates.

Based on empirical measurements, the energy of NRKW calls is more homogeneously distributed across frequencies than SRKW calls (i.e., decrease of −0.27 dB per 300 Hz band for NRKW vs −0.66 dB per 300 Hz band for SRKW, Fig 2). For SRKW, because most of the energy is concentrated at low frequencies, low frequency components of the calls can be detected at farther ranges than the high frequency components that are attenuated rapidly. The NRKW call detection ranges were modeled in an environment where ocean ambient sound is more pronounced at low frequencies (Juan de Fuca Strait off Port Renfrew), and since the energy of NRKW calls is spread relatively evenly across frequencies, it means that the low-frequency components of their calls are quickly masked. As a result, the higher-frequency components of killer whale calls, which fall outside the primary masking range, are detected at greater distances. Yurk et al. [41] suggested that calls may have adapted design features, such as two-voice calling to reduce masking. For example, two-voice calling is a strategy used by Emperor penguins to find their mate after returning to a noisy colony [79]. The sensitivity analysis conducted showed that the frequency distribution of source levels and the ambient sound levels were the two factors that have the greatest influence on the detection range modeling. Consequently, it is important to carefully characterize these two factors to obtain the most realistic detection range estimates. It is important to set the maximal detection frequency of the automated detector high enough to capture both the low and high frequency components of calls.

It is relatively straightforward to estimate ambient sound levels using *in situ* measurements. Here we used sound pressure levels from recordings collected by calibrated instruments. We used sound pressure levels averaged over 1 minute as it is becoming a common time resolution for soundscape analysis [80–82] The sensitivity analysis showed that using shorter averaging windows does not affect the outcome of the detection range modeling substantially (S3 Table).

Using as much data as possible helps to accurately represent the variations in ambient sound conditions at each study site and provides more reliable detection range estimates. We used at least 10,000 ambient sound measurements for each modeling scenario. Because of the limited data available, we could only use 5,000 ambient sound measurements for Studies Bay in winter. Consequently, results from this modeling scenario should be interpreted with caution. The frequency distribution of source levels is the most sensitive parameter for the detection range modeling. It is also the most difficult to estimate. Source levels of killer whale calls (or whale calls in general) are typically only reported as normally distributed values (characterized by their mean and standard deviation) covering all frequencies of the call (i.e., broadband). This, however, rarely represents the source level distribution in broadband cetacean signals. Using the mean and standard deviation of the broadband source level does not accurately represent the dynamics of whale call detection ranges and more importantly may often overestimate the detection range of these calls under noisy conditions. This is a serious problem when the goal of detection is to track whales to mitigate disturbances.

Here, to estimate the relative frequency distribution of source levels, we used measurements of killer whale calls with high SNR, assuming that the animals were close to the instrument and that the acoustic attenuation was negligeable. Based on local knowledge of the area, we believe that this is a realistic assumption. However, performing acoustic localization and backpropagating received levels would be preferrable (when possible). Also, it is important to gather a comprehensive sample of calls preferably collected at various locations with different ambient sound levels. This is often difficult to achieve as one also requires calls with high SNR. The potential effect of the dual components of killer whale calls on the detection range could not be fully assessed in this study because of lack of sufficient examples from different environments. It would need to be considered, however, when detection range is modeled in quieter locations. The dual

component influence on the source levels may explain the lower levels considered for NRKW calls in this study compared to those reported elsewhere [27]. One of the important results of this study, however, is the demonstration of an influence of source levels in different frequency bands on the estimated call detection rate. In order to improve the reliability of detection range estimates, we encourage researchers to report source level measurements not only as a broadband value but always in different frequency bands. Vocalization depth distribution and the difference in propagation loss did not substantially influence the outcome of the detection range modeling. While there are differences in propagation loss values between summer and winter, the differences occur mainly at long ranges (beyond several kilometers). Consequently, the median detection ranges are not affected appreciably by these seasonal differences of the propagation loss.

The approach used in this paper is designed to compute the maximum detection range for detecting a single vocalizing whale.. If many whales are co-located and vocalizing together, then the probability of acoustic detection is expected to increase due to the increased volume of calls produced at different distances from the receiving hydrophone (i.e., increased probability of detecting calling animals from farther away). All detection range probabilities presented were estimated using a detection threshold of 5 dB (Eq 2). This parameter is dependent on the detection process used and will vary depending on the monitoring scenario. If all acoustic data are manually annotated by an analyst, the detection threshold would likely be closer to 0 dB. The performance of an automated detector, however, needs to be quantified by signal-to-noise ratio intervals in order to select the appropriate detection threshold value [75]. One needs to define a satisfactory precision and recall level depending on the purpose of the system, e.g., whale tracking via a network of PAM systems located at various geographic locations to alert vessel operators of the presence of whales. Results from our study suggest that a network of PAM stations will be most effective at detecting the presence of killer whales (including SRKW) during the summer, and better at some locations than others. While the proposed approach is focusing on the detectability of killer whales, it can be easily generalized to other species by adapting the dive profiles (if tag data are available) and using an appropriate propagation loss model. The proposed approach could also assist when designing a passive acoustic monitoring plan for mitigation of noise exposure during commercial activities that produce high sound levels by determining PAM locations that maximize detection ranges at times when the activities occur. In critical areas, using multiple synchronized hydrophones would allow to perform beamforming and add an array gain to the Eq 1 which would substantially increase the detection range [49,83]. Note that future work will include field studies to ground truth the detection range estimates from the modeling with empirical *in situ* measurements. Initial results of this analysis demonstrated the importance of understanding the frequency distribution of source levels [84]. We would encourage future studies to report source levels in different frequency bands rather than only broadband values.

While this was not the primary focus of this study, the probability of on-axis detection estimated here can provide valuable information for animal density estimation using passive acoustics [85]. As noted previously, the low-frequency components of killer whale pulsed calls are omnidirectional and generally determine the maximum detection range. This was the case for the vast majority of SRKW simulations we conducted (S12 Fig). In such cases, the estimated probability of on-axis detection is very close to the true probability of acoustic detection and could be used directly in density estimation studies employing distance sampling or Spatially Explicit Capture–Recapture approaches based on single calls. However, as we found (primarily for NRKW), the higher-frequency, more directional components of pulsed calls can sometimes set the maximum detection range (S13 Fig). In these cases, directionality patterns would need to be defined (either with *in situ* measurements or using modeling) and directionality would need to be incorporated into the analysis using methods similar to those described by [72] before the estimates could be used for density estimation.

## Conclusion

Estimating the maximum detection range of marine mammal sounds is important for designing and interpreting results from passive acoustic monitoring studies and when designing monitoring networks to track whales acoustically. Yet, it is not often characterized. In this study we estimated the detection range of stereotypic pulsed calls from Southern and

NRKW at several passive monitoring stations deployed by Fisheries and Ocean Canada in waters off the British Columbia coast. The detection range was estimated using ambient sound levels measured *in situ*, modeled propagation losses determined by the acoustic model MONM and applied Monte Carlo simulations to capture the variability in call source level and animal depth. The approach presented shows the probability of detecting killer whale calls at each location and also identifies which frequency bands are most important for detection over long distances. In most cases, the omnidirectional low-frequency components determine the maximum detection range, allowing us to estimate the true probability of acoustic detection. In some cases, however, the more directional high-frequency components can become important for the maximum detection range, in which case the estimated probability of detection applies only to calls received on-axis. We show that the distribution of the source level of calls across frequencies in conjunction with the ambient sound level are principal factors that affect detection ranges, but are rarely reported

## Supporting information

**S1 Table. End points of model transects originating from each model site.**
(DOCX)

**S2 Table. Geoacoustic profiles applied at the model sites.** Within each depth range, each parameter varies linearly within the stated range.
(TIF)

**S3 Table. Sources of acoustic data used for ambient sound level measurements at each of the model sites, and for each modeled season.**
(DOCX)

**S4 Table. Median detection ranges (in meters) for probability of on-axis detection (P) of 0.1, 0.5 and 0.9 for Monte Carlo simulations run using either 10 second or 60 second data windows to compute the ambient noise levels (computed for 1 week of ambient data).**
(TIF)

**S1 Fig. Example of killer whale pulsed call recording used to determine the energy distribution across frequency. Yellow boxes indicate annotations used to determine the frequency distributions of the calls.**
(PNG)

**S2 Fig. Geological Survey of Canada (GSC) and the BC Marine Ecological Classification (BCMEC) geological map of the western British Columbian coast.**
(TIF)

**S3 Fig. Sound speed profiles used to model winter (blue) and summer (red) conditions at each of the model sites.**
(TIF)

**S4 Fig. Distribution of broadband source levels sampled in the Monte Carlo process for SRKW call detection range calculations. For (left) summer and (right) winter background conditions at (top to bottom) East Point, Enterprise Reef, Mouat Point, Port Renfrew, Sheringham Point, Sturdies Bay, Swiftsure Bank, and Tilly Point.**
(TIF)

**S5 Fig. Distribution of broadband source levels sampled in the Monte Carlo process for NRKW call detection range calculations. For (left) summer and (right) winter background conditions at (top to bottom) Port Renfrew, Sheringham Point, and Swiftsure Bank.**
(TIF)

**S6 Fig. Statistical distribution of source levels in 300 Hz bands from the Monte Carlo process for SRKW call detection range calculations, as a function of frequency.** For (left) summer and (right) winter background conditions at (top to bottom) East Point, Enterprise Reef, Mouat Point, Port Renfrew, Sheringham Point, Sturdies Bay, Swiftsure Bank, and Tilly Point.
(TIF)

**S7 Fig. Statistical distribution of source levels in 300 Hz bands from the Monte Carlo process for NRKW call detection range calculations, as a function of frequency.** For (left) summer and (right) winter background conditions at (top to bottom) Port Renfrew, Sheringham Point, and Swiftsure Bank.
(TIF)

**S8 Fig. Distribution of vocalization depths sampled in the Monte Carlo process for SRKW call detection range calculations, as a function of frequency.** For (left) summer and (right) winter background conditions at (top to bottom) East Point, Enterprise Reef, Mouat Point, Port Renfrew, Sheringham Point, Sturdies Bay, Swiftsure Bank, and Tilly Point.
(TIF)

**S9 Fig. Distribution of vocalization depths sampled in the Monte Carlo process for NRKW call detection range calculations, as a function of frequency.** For (left) summer and (right) winter background conditions at (top to bottom) Port Renfrew, Sheringham Point, and Swiftsure Bank.
(TIF)

**S10 Fig. Percentile distribution of ambient sound levels (in 300 Hz bands) from the recorded data samples used in the Monte Carlo process for SRKW call detection range calculations, as a function of frequency.** For (left) summer and (right) winter background conditions at (top to bottom) East Point, Enterprise Reef, Mouat Point, Port Renfrew, Sheringham Point, Sturdies Bay, Swiftsure Bank, and Tilly Point.
(TIF)

**S11 Fig. Percentile distribution of ambient sound levels (in 300 Hz bands) from the recorded data samples used in the Monte Carlo process for NRKW call detection range calculations, as a function of frequency.** For (left) summer and (right) winter background conditions at (top to bottom) Port Renfrew, Sheringham Point, and Swiftsure Bank.
(TIF)

**S12 Fig. Percentage of the results for which each frequency band yielded the maximum detection range in the Monte Carlo process for SRKW call detection range calculations, as a function of frequency.** For (left) summer and (right) winter background conditions at (top to bottom) East Point, Enterprise Reef, Mouat Point, Port Renfrew, Sheringham Point, Sturdies Bay, Swiftsure Bank, and Tilly Point.
(TIF)

**S13 Fig. Percentage of the results for which each frequency band yielded the maximum detection range in the Monte Carlo process for NRKW call detection range calculations, as a function of frequency.** For (left) summer and (right) winter background conditions at (top to bottom) Port Renfrew, Sheringham Point, and Swiftsure Bank.
(TIF)

**S14 Fig. Propagation loss modeled in the 1900–2200 Hz band at a-b) East Point, c-d) Enterprise Reef, e-f) Mouat Point, and g-h) Port Renfrew, for winter (left column) and summer (right column) conditions, as a function of range and depth along a single radial.**
(TIF)

**S15 Fig. Propagation loss modeled in the 1900–2200 Hz band at a-b) Sheringham Point, c-d) Sturdies Bay, e-f) Swiftsure Bank, and g-h) Tilly Point, for winter (left column) and summer (right column) conditions, as a function of range and depth along a single radial.**
(TIF)

**S16 Fig. a) Negatively slopped and b) uniform frequency distribution of NRKW source levels used for the sensitivity analysis. Statistical distribution of source levels (in 300 Hz bands) from the Monte Carlo simulation, as a function of frequency.**
(TIF)

**S17 Fig. a) Winter and b) summer ambient levels at Port Renfrew used for the sensitivity analysis. Percentile distribution of background sound levels (in 300 Hz bands).**
(TIF)

**S18 Fig. a) Uniform and b) log-logistic depth distributions used for the sensitivity analysis.**
(TIF)

## Acknowledgments

We would like to thank Brianna Wright (Fisheries and Oceans Canada, DFO) for providing animal depth data from DTAGs, Svein Vagle (DFO) and Caitlin O'Neil (DFO) for providing the acoustic data from Swiftsure Bank and Port Renfrew, Paul Cottrell (DFO) for providing the acoustic data from the Whale Tracking Network operated by DFO Marine Mammal Management, and the Vancouver Fraser Port Authority and Ocean Networks Canada for supporting the data acquisition at East Point. Northern Resident Killer Whale calls used in this study were collected by Jennifer Wladichuk (JASCO Applied Sciences) with field support from Svein Vagle, Caitlin O'Neill and Peter Van Buren (DFO), Jens Koblitz (Max Planck Institute of Animal Behavior), and Marie Zahn (University of Washington).

## Author contributions

**Conceptualization:** Xavier Mouy, Harald Yurk.

**Data curation:** Xavier Mouy, Melanie Austin, Jennifer Wladichuk.

**Formal analysis:** Xavier Mouy, Melanie Austin.

**Funding acquisition:** Melanie Austin, Harald Yurk.

**Investigation:** Xavier Mouy, Melanie Austin, Harald Yurk.

**Methodology:** Xavier Mouy, Melanie Austin.

**Project administration:** Melanie Austin.

**Resources:** Jennifer Wladichuk.

**Software:** Xavier Mouy, Melanie Austin.

**Supervision:** Harald Yurk.

**Validation:** Xavier Mouy, Melanie Austin.

**Visualization:** Xavier Mouy, Melanie Austin.

**Writing – original draft:** Xavier Mouy, Melanie Austin, Harald Yurk.

**Writing – review & editing:** Xavier Mouy, Melanie Austin, Jennifer Wladichuk, Harald Yurk.

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
