## [Decision Letter · Decision Letter 0]

18 Feb 2025

Dear Dr. Mouy,

Thank you for submitting your manuscript to PLOS ONE. After careful consideration, we feel that it has merit but does not fully meet PLOS ONE’s publication criteria as it currently stands. Therefore, we invite you to submit a revised version of the manuscript that addresses the points raised during the review process.

We look forward to receiving your revised manuscript.

Kind regards,

Vitor Hugo Rodrigues Paiva, Ph.D.

Academic Editor

PLOS ONE

Journal Requirements:

3. Thank you for stating the following financial disclosure: This study was funded by Fisheries and Oceans Canada and JASCO Applied Sciences. XM also received support from the Investment in Science Fund at the Woods Hole Oceanographic Institution.

4. Thank you for stating the following in the Acknowledgments Section of your manuscript: We would like to thank Brianna Wright (Fisheries and Oceans Canada, DFO) for providing animal depth data from DTAGs, Svein Vagle (DFO) and Caitlin O’Neil (DFO) for providing the acoustic data from Swiftsure Bank and Port Renfrew, Paul Cottrell (DFO) for providing the acoustic data from the Whale Tracking Network operated by DFO Marine Mammal Management, and the Vancouver Fraser Port Authority and Ocean Networks Canada for supporting the data acquisition at East Point. Northern Resident Killer Whale calls used in this study were collected by Jennifer Wladichuk (JASCO Applied Sciences) with field support from Jens Koblitz (Max Planck Institute of Animal Behavior), Peter Van Buren (DFO), and Marie Zahn (University of Washington). This study was funded by Fisheries and Oceans Canada and JASCO Applied Science. XM also received support from the Investment in Science Fund at the Woods Hole Oceanographic Institution.

Please remove any funding-related text from the manuscript and let us know how you would like to update your Funding Statement. Currently, your Funding Statement reads as follows: This study was funded by Fisheries and Oceans Canada and JASCO Applied Sciences. XM also received support from the Investment in Science Fund at the Woods Hole Oceanographic Institution.

5. Please update your competing interest statement.

Reviewers' comments:

Reviewer's Responses to Questions

**Comments to the Author**

1. Is the manuscript technically sound, and do the data support the conclusions?

Reviewer #1: Yes

Reviewer #2: No

2. Has the statistical analysis been performed appropriately and rigorously?

Reviewer #1: Yes

Reviewer #2: Yes

3. Have the authors made all data underlying the findings in their manuscript fully available?

Reviewer #1: Yes

Reviewer #2: No

4. Is the manuscript presented in an intelligible fashion and written in standard English?

Reviewer #1: Yes

Reviewer #2: Yes

Reviewer #1: Review of PLOS ONE Manuscript: "Modeling the detection range of pulsed calls from resident killer whale in nearshore waters of British Columbia, Canada"

Summary and Significance

The manuscript presents a Monte Carlo approach to modeling the detection range of pulsed calls from resident killer whales (Orcinus orca) in the Salish Sea using passive acoustic monitoring (PAM). The study incorporates empirical ambient noise data, acoustic propagation modeling, and variability in call source levels and animal depths to estimate detection probabilities at various locations. The results provide valuable insights into the efficacy of PAM for monitoring Southern and Northern Resident Killer Whales. This research is particularly relevant for conservation and management efforts, given the increasing anthropogenic noise in marine environments and the necessity for effective whale monitoring and mitigation programs.

Assessment of Claims and Contextualization

The manuscript effectively places its claims in the context of previous literature on PAM, detection range modeling, and marine mammal monitoring. The introduction comprehensively reviews the significance of whale vocalizations and previous methodologies employed for estimating detection ranges. The study also contextualizes the challenges posed by noise pollution and seasonal variations in the Salish Sea. However, the discussion could benefit from a more explicit comparison with other detection range studies, particularly those that have used alternative modeling techniques or empirical validation methods.

Validity of Data and Analyses

The methodology is generally sound, and the use of Monte Carlo simulations provides a robust probabilistic approach to estimating detection range variability. The study appropriately accounts for variability in source levels and ambient noise conditions. However, the manuscript would benefit from further clarification on:

• The assumptions and limitations of using specific source level distributions.

• Potential biases introduced by the limited number of high-SNR calls used to estimate frequency-dependent source levels.

• The potential effects of Lombard responses (i.e., call amplitude modulation in response to noise) on detection range estimates.

• Additional text and plot in the results to show probability of detection under different ambient noise levels.

Reproducibility and Methodological Transparency

The manuscript provides sufficient methodological details to allow reproduction of the study. The authors have made relevant data available via an open repository, adhering to PLOS ONE’s data-sharing policies. However, certain aspects of the methodology could be clarified further:

• A more detailed description of how high SNR calls were annotated, e.g. individual calls, call bouts, any buffer before and after call, etc. and state explicitly over what time domain the SL calculation was done.

• A clearer description of the depth distribution data used, e.g. what type of tag was used, were these calling depths or just general depth ranges?

• Additional discussion on whether the source level distributions account for directional biases in killer whale vocalizations.

Clarity and Organization

The manuscript is well-structured and very well written with clearly defined sections and logical flow. The results are presented effectively with comprehensive figures and tables.

• Suggest that Figure 9 have standardized y-axes on all plots and increased resolution.

Requested Revisions and Edits

1. Methodological Clarity:

o A more detailed description of how high SNR calls were annotated, e.g. individual calls, call bouts, any buffer before and after call, etc. and state explicitly over what time domain the SL calculation was done.

o A clearer description of the depth distribution data used, e.g. what type of tag was used, were these calling depths or just general depth ranges?

2. Discussion Section:

o Strengthen the discussion by explicitly comparing detection range results with those from other studies employing different methodologies (e.g., empirical validation vs. model-based approaches).

o Address potential gaps in source level distribution assumptions and their implications on detection range estimates.

o The authors miss an opportunity to discuss how this approach could be used in Density Estimation from PAM recorders given we know cue counts (from tag data).

o Add more text on how Lombard effects (e.g. if killer whales change their call amplitudes in response to noise) could affect detection range estimates. Presumably this could be simulated and while I wouldn’t request that for this study it could be noted as a future direction for the work.

o Discuss whether directional characteristics of killer whale calls might impact estimated detection probabilities.

o Add discussion of how this approach may be used in Density Estimation from PAM. This may also be a useful addition to the Introduction.

I would recommend that this manuscript be published after edits and comments are addressed.

Minor text edits and comments can be found on the attached document.

Reviewer #2: Overall

This is a useful paper and with practical application for acoustic monitoring of killer whales, however, there are some fundamental issues with the methods along with more minor errors and oversights that which need addressed.

My main issue with the paper is that that the authors are not calculating the probability of detection. They are calculating the probability of a detecting a killer whale pulsed call for a killer whale that happens to be facing the hydrophone at a mean source level over a time series of noise. In the paper, the authors are often careful to point out that they are measuring maximum detection range but regularly use terms like “probability of detection” that would generally be associated with a true detection function used in distance sampling. The problem with measuring maximum range is that it is not particularly useful because at that range you will always miss most animals unless they produce sounds that are omni-directional. In this case, the authors are measuring killer whale pulsed calls which are produced via the same mechanism as clicks and are thought to be directional, especially at higher frequencies which are included in the simulation (Miller et al, 2002).

The authors use equation 3 (which is incorrect, but I assume you used correctly in the simulations) to calculate the maximum detection range. They then use a Monte Carlo simulation to calculate the uncertainty in the maximum range which they term the probability of detection with range for calls. This is not the probability of detection as the term is generally used, it is the probability of receiving a call at the mean source level for a killer whale facing the hydrophone. However, the authors have done all the hard work in getting together good quality inputs for a true probability of detection simulation and so it seems a shame to not take this further.

My advice would be to redo the analysis and use a Monte Carlo approach to simulate the probability of detecting a whale as in (e.g. Frasier et al, 2016). These methods would allow directionality to be incorporated and utilise all the very good work in this paper on propagation, noise level estimation and explore the frequency dependence of source levels. Note that the probability of detection simulation approach uses distributions like source levels, depth and vertical angle (which you can get from tag data) along with beam profile to calculate a probability of detecting a sound with range. The uncertainty in the inputs and input distributions (rather than the distributions themselves) is used to calculate an error in the simulation rather than what the authors have done which is to take empirical distributions of source levels etc. and use them to calculate an uncertainty in the maximum detection range with distance. The probability of detection can be mapped to noise levels and then an average probability of detection over a time series of noise calculated. This can be used to determine an effective detection radius which can then be plotted on Figure 6 along with maximum detection range. This would make the paper much more useful and prevent the misuse of results (i.e. by folk who think this represent the probability of detecting a killer whale).

More generally it would be good in the results section to provide some plots showing time long term time series of the change in maximum range and also to explore which frequency bands are resulting in the maximum ranges as I imagine higher frequencies have higher absorption, but noise is higher at lower frequencies. Is there a portion of the call that is usually detected more easily or is this quite dependent on environmental factors?– that isn’t clear but all the work has been done from equation 4.

What I am proposing (and hoping for) is a significant rewrite and re-analysis and as such this constitutes major revisions. I would be happy to review the paper again if this were done, however, I understand that this may not be possible and there is still value in what has been done here. If the authors rejected these suggestions, then this version of the paper could be published if the specific issues below are rectified. Particularly, there needs to be careful attention to ensure that there is no ambiguity about what is termed “probability of detection” and some of the above points need to be added to the discussion.

I would also note that all figures are blurry in the pdf which I assume is not be the fault of the authors but is frustrating for a reviewer.

Specifics

Line 54 and it provides information when animals are underwater.

Line 54 I know what you mean but PAM is not only useful for animal sounds – it is useful for a whole host of different things – noise etc. Would reword.

Line 70 Sudden switch from passive to active text. Would keep it consistent.

Line 100. Does this account for absorption? That needs to be mentioned.

Line 106 – yeah but it’s an arbitrary threshold anyway so mentioning processing gains seems a little arbitrary? Would maybe just remove that.

Line 110 Equation is wrong. In Eq 1 you have used PL as a transmission loss and it has units of dB re 1 m2. In eq 3 you have used PL as a coefficient i.e TL = PL*log10(R). The coefficient is dimensionless. Also, you have described PL in Eq 6 which is inconsistent with equation 4.

Line 199 Equation 6 should be base ten log?

Line 352 should maybe be average detection range (change throughout where appropriate)

Line 376 – so does this mean your icListen systems were not calibrated (not mentioned in methods)? If this is true, then you need to discuss somewhere why you think this not an issue – e.g. the manufacturer’s quoted sensitivities are known to be quite accurate. A SoundTrap sensitivity varies by 6dB which will have significant effects on your range estimation – you are not using these devices but icListen systems are similar…ish. If calibration is an issue then this should be added to the sensitivity analysis.

Line 439 No it is not a calculation of the probability of detecting a single vocalisation – see discussion above.

Line 446 Yes, the language “detection range probabilities” is correct (probably should be “maximum detection range probabilities”) – use this above.

Line 466. Again, I don’t think maximum detection range is a particularly useful parameter (although frequently reported in ltrature) for any directional vocalization. Nice to know but not that useful for say density estimation. For example, if simulating the probability of detection, then your probability scales with whatever you set as the area you are monitoring over - and thus the area term in the density estimation equation cancels out. Similarly, many experimentally defined detection probabilities will have a maximum detection range quoted.

Line 472. Remove “probability of detection”.

References

Miller, P. (2002) ‘Mixed-directionality of killer whale stereotyped calls: a direction of movement cue?’, Behavioral Ecology and Sociobiology, 52(3), pp. 262–270. Available at: https://doi.org/10.1007/s00265-002-0508-9.

Frasier, K.E. et al. (2016) ‘Delphinid echolocation click detection probability on near-seafloor sensors’, The Journal of the Acoustical Society of America, 140(3), pp. 1918–1930. Available at: https://doi.org/10.1121/1.4962279.

**Do you want your identity to be public for this peer review?** For information about this choice, including consent withdrawal, please see our Privacy Policy

Reviewer #1: No

Reviewer #2: No

---

## [Author Response · Author response to Decision Letter 1]

14 Jun 2025

All comments from the editor and the reviewers have been addressed and are in the "Response to Reviewers" document

---

## [Decision Letter · Decision Letter 1]

30 Jul 2025

Dear Dr. Mouy,

Thank you for submitting your manuscript to PLOS ONE. After careful consideration, we feel that it has merit but does not fully meet PLOS ONE’s publication criteria as it currently stands. Therefore, we invite you to submit a revised version of the manuscript that addresses the points raised during the review process.

We look forward to receiving your revised manuscript.

Kind regards,

Vitor Hugo Rodrigues Paiva, Ph.D.

Academic Editor

PLOS ONE

Journal Requirements:

Reviewers' comments:

Reviewer's Responses to Questions

**Comments to the Author**

Reviewer #2: (No Response)

Reviewer #3: (No Response)

Reviewer #4: (No Response)

2. Is the manuscript technically sound, and do the data support the conclusions?

Reviewer #2: Yes

Reviewer #3: Yes

Reviewer #4: Yes

3. Has the statistical analysis been performed appropriately and rigorously?

Reviewer #2: Yes

Reviewer #3: Yes

Reviewer #4: Yes

4. Have the authors made all data underlying the findings in their manuscript fully available?

Reviewer #2: Yes

Reviewer #3: Yes

Reviewer #4: Yes

5. Is the manuscript presented in an intelligible fashion and written in standard English?

Reviewer #2: Yes

Reviewer #3: Yes

Reviewer #4: Yes

Reviewer #2: The authors should be commended on responding carefully to the comments of both reviewers, however, there are still some issues here.

Overall, the reviewers have fixed the immediate errors in the manuscript and made constructive changes to all sections based on the reviewer's comments.

However, there is still an issue here around the term probability of detection. I note the authors response and in, a way, agree with all of it. The low frequencies will be nearer omni- directional, the high frequencies will be directional. I also note that I did not myself clear in the review when saying “facing the hydrophone at a mean source level” and I accept the authors have gone to great lengths to accurately assess source level distributions for different frequencies (I was thinking of beam loss but that was certainly not clear in the review text).

The authors have a distribution of source levels, a distribution of depths, noise measurements and conducted advanced propagation modelling all with respect different frequency bands. For a low frequency sound, they have essentially simulated a true probability of detection because the sounds are near omni-directional and therefore direction of the animal is not important. For the higher frequency sounds, they have estimated maximum on-axis detection range (i.e. when an animal is pointing directly at a hydrophone), as beam profile and directionality need to be taken into account for a true probability of detection estimation. The authors clearly understand this from the response to the review, and they have done a good job in explaining what the study is about in the introduction regards maximum detection ranges. In the previous review, I suggested a large re-analysis which I understood probably would not be possible due to time constraints and suggested that if that was not possible, they disambiguate the term “probability of detection” and update the manuscript with fixes etc (which they have done). The specific response to by the authors was.

“We have addressed the comments by clarifying the goal of the study and the underlying

methodology. We do not believe that a re-analysis of the same data will produce results that are fundamentally different from the ones we presented because directionality was not considered and is only relevant to the propagation of HF parts of the calls that attenuate much faster than the LF parts”

except that Figure 9 and L433 L462 clearly indicate that higher frequency calls are responsible for the larger detection ranges in some cases. So, if referring to a simulation which takes into account directionality, the statement, “We do not believe that a re-analysis of the same data will produce results that are fundamentally different from the ones we presented” maybe would turn out to be true but surely cannot be backed up unless I’ve misunderstood something here?

Aside from that, the authors have decided to move to the terminology “probability of acoustic detection” which really should be “Probability of possibility of acoustic detection” (has a nice ring to it right?). I think that using “probability of on-axis detection” is probably a good compromise (even though in the LF case you will be looking at close to a true probability of detection) and I would like to see this in all plots and all references to the probability of detection or similar. For example, statements like L537 need to be changed

“The approach used in this paper is designed to compute the probability of detecting a single vocalizing whale” needs to be changed to “The approach used in this paper is designed to compute the maximum detection range for detecting a single vocalising whale.”

And Line 573

“The approach presented shows the probability of detecting killer whale

calls at each location” needs to be changed to “The approach presented shows the probability of detecting killer whale on-axis calls at each location”.

Finally, both reviewers asked for further discussion of density estimation. Whilst I am happy for pushback on some of the points responded to, I think the paper has to have a paragraph on this. In particular, the authors need to discuss the points above about how the low frequencies will be close to a true probability of detection and the high frequency cases not so much. The author response to reviewer 1

“While this is a worthwhile goal and should be addressed in future study, we feel that this study does not provide enough relevant information on the subject of density estimations using PAM, because it focusses on the detection ranges of single calls. Killer whales like many other delphinids are highly vociferous and a study on density estimation should focus on that aspect of the vocal behavior.”

I don’t think is adequate. Density estimation is often performed on single calls via distance sampling or SECR of which a probability of detection of a single call is a key component. You have come to close to calculating a probability of detection (when most calls that contribute to probability of on-axis detection are low frequency) but the issue with directionality and HF call detection is the reason this cannot be used for density estimation. In fact, as I mentioned in my first review, you have come so close to calculating the probability of detection it’s a shame not to go the whole way (although again I fully understand time constraints).

I understand this may seem a little negative, however, I think this is an impressive paper, useful paper and a credit to the authors. The reason I am insisting on the above changes, is because I am worried this paper will be misused. For example, someone taking the probability of detection and using to estimate density from single sensors.

So in summary

• Change probability of detection to something unambiguous like “probability of on-axis detection” in all references of probability of detection, including figures.

• Add a discussion paragraph on density estimation specifically with the points discussed above.

Once that is done, this paper is ready to go.

Reviewer #3: Great article, just couple minor comments-

- The last sentence of the abstract is rather vague make more infromative how your findings help future studies also make more explicit in conclusion

- Discussion should be more concise and instructive

- I think you should mention the main differences between towed and moored hydrophone surveys and also mention that its optimal to have visual surveys conducted simultaneously (cite - https://doi.org/10.1111/mms.13113)

- You should briefly outline the functional differences between clicks, whistles and pulsed calls for biologists who are not familiar with cetacean acoustics - PlosOne reaches a wide audience.

- It would be good to include a paragraph in the discussion about future possibilities to use AI/ML tools to assist in Killer Whale Acoustic detection and mention advances in this field for other species i.e., https://www.projectceti.org/

Reviewer #4: General comments:

The paper is generally well written and presents results from modeling the detection range of pulsed calls from a resident killer whale population off the coast of British Columbia, Canada. I believe it merits eventual publishing, but it would benefit from some revisions.

I suggest providing additional context regarding the acoustic recordings used in the study. In particular, information about the depth of the hydrophones and the time window over which recordings were made is currently missing or not clearly stated. Additionally, several figures would benefit from minor adjustments to improve clarity, for example adding labels directly on plots to avoid frequent reference to captions. Finally, the Discussion section would benefit from a few clarifications. Addressing these points would enhance the overall quality and interpretability of the manuscript.

Introduction:

- Lines 64-67: It would be relevant to provide a bit more detail on the acoustic characteristics of the vocalizations mentioned (typical frequency ranges and durations).

- Line 91: You should specify the typical values for this narrow frequency band to provide more context.

- Line 94: Same comment but for the wide frequency band.

Methods:

Approach:

- Line 125: Could you explain the reason for choosing 5 dB as DT?

Source Level Estimation:

- Lines 152-156: This description of call types does not belong in the Methods. I recommend moving it to the Introduction, where it would fit as background information (see my comment on lines 64–67).

Fig 2:

You should label “SRKW” and “NRKW” directly on top of the spectrogram panels so that we can immediately understand which column corresponds to which group, without needing to refer back to the caption.

Vocalization depth:

- Line 207: I don’t think the depth of the hydrophones was mentioned earlier in the Methods. It would be helpful to include a brief section/paragraph describing the deployment details of the instruments, including their depth and the time period over which recordings were made.

Propagation loss modeling:

Fig 4:

You should specify in the caption that the red lines correspond to the transects.

Results:

Fig 5:

You should label “winter” and “summer” directly on top of the plots.

Fig 6:

You should specify blue for summer and orange for winter in the caption.

Fig 9:

You should add labels directly on the plots to ensure consistency with previous figures and to make the figure easier to interpret without having to refer to the caption each time.

Fig 10:

Same comment as for Fig 9.

Fig 11:

Same comment as for Fig 9.

Discussion:

- Lines 400-401: This sentence appears to contradict line 391. How can Mouat Point be listed among the locations with larger detection ranges and also be described as having one of the shortest median detection ranges? Please clarify this point.

- Lines 421-424: The explanation that shorter summer detection ranges may be due to "under sampling" of winter data seems counterintuitive. If winter data are limited, it is unclear how this would explain longer detection ranges in winter than in summer. Please clarify the logic here.

- Line 422: “As already indicated”: this is slightly vague. Perhaps refer back more explicitly to the earlier section where this was discussed.

Fig 13:

Same comment as for Fig 9.

- Lines 504-512: Please clarify whether the model represents a minimum or maximum expected detection performance and in what context.

References:

Line 592: The citation is missing.

**Do you want your identity to be public for this peer review?** For information about this choice, including consent withdrawal, please see our Privacy Policy

Reviewer #2: No

Reviewer #3: **Yes: ** Isla Duporge

Reviewer #4: **Yes: ** Alexandra Nathalie Constaratas

---

## [Author Response · Author response to Decision Letter 2]

16 Aug 2025

The detailed response to reviewers is in the MS Word document provided (Response to Reviewers.docx)

---

## [Editor Report · Decision Letter 2]

25 Aug 2025

Modeling the detection range of pulsed calls from resident killer whale in nearshore waters of British Columbia, Canada.

PONE-D-24-60689R2

Dear Dr. Mouy,

We’re pleased to inform you that your manuscript has been judged scientifically suitable for publication and will be formally accepted for publication once it meets all outstanding technical requirements.

Kind regards,

Vitor Hugo Rodrigues Paiva, Ph.D.

Academic Editor

PLOS ONE
---

## [Editor Report · Acceptance letter]

PONE-D-24-60689R2

PLOS ONE

Dear Dr. Mouy,

I'm pleased to inform you that your manuscript has been deemed suitable for publication in PLOS ONE. Congratulations! Your manuscript is now being handed over to our production team.

Kind regards,

on behalf of

Dr. Vitor Hugo Rodrigues Paiva

Academic Editor

PLOS ONE